# Targeting the WSB2–NOXA axis in cancer cells for enhanced sensitivity to BCL-2 family protein inhibitors

**Dongyue Jiao[1†], Kun Chang[2,3†], Jiamin Jin[1], Yingji Chen[1], Mo Ren[4], Yucong Zhang[5], Kun Gao[6,7], Yaoting Xu[8*‡], Lixin Wang[5*‡], Chenji Wang[1*‡]**

[1]State Key Laboratory of Genetics and Development of Complex Phenotypes, Shanghai Stomatological Hospital and School of Stomatology, MOE Engineering Research Center of Gene Technology, Shanghai Engineering Research Center of Industrial Microorganisms, School of Life Sciences, Fudan University, Shanghai, China; [2]Department of Urology, Fudan University Shanghai Cancer Center, Shanghai, China; [3]Department of Oncology, Shanghai Medical College, Fudan University, Shanghai, China; [4]Department of Urology, Inner Mongolia Urological Institute, Inner Mongolia Autonomous Region People's Hospital, Inner Mongolia, China; [5]Department of Vascular Surgery, Zhongshan Hospital, Fudan University, Shanghai, China; [6]Department of Clinical Laboratory, Shanghai First Maternity and Infant Hospital, School of Medicine, Tongji University, Shanghai, China; [7]Shanghai Key Laboratory of Maternal and Fetal Medicine, Shanghai First Maternity and Infant Hospital, Shanghai, China; [8]Department of Urology, Shanghai Fourth People's Hospital Affiliated to Tongji University School of Medicine, Shanghai, China

***For correspondence:**
2000019@tongji.edu.cn (YX);
wang.lixin@zs-hospital.sh.cn (LX);
Chenjiwang@fudan.edu.cn (CW)

[†]These authors contributed equally to this work
[‡]These authors also contributed equally to this work

**Competing interest:** The authors declare that no competing interests exist.

## eLife Assessment

This study reports a **fundamental** observation concerning cell death regulation by the anti-apoptotic BCL2 family NOXA. The authors **convincingly** demonstrate that NOXA is destabilized through the interaction with WSB2, a substrate receptor in CRL5 ubiquitin ligase complex, sensitizing the cells to treatments. These are key findings for cell biologists and cancer researchers as they identified a new target impacting drug responsiveness in cancer therapies.

**Abstract** Anti-apoptotic B-cell lymphoma-2 (BCL-2) family proteins are frequently overexpressed in various cancers, playing a pivotal role in cancer initiation and progression, as well as intrinsic or acquired resistance to therapy. Although inhibitors targeting BCL-2, such as Venetoclax, have shown efficacy in hematological malignancies, their therapeutic potential in solid tumors remains limited. Identifying novel molecular targets to overcome resistance to these inhibitors is of significant clinical importance. Here, we provide evidence of a strong synthetic lethality between WSB2, a previously underexplored substrate-binding receptor of the Cullin 5–RBX2–Elongin B/C (CRL5) E3 ubiquitin ligase complex, and anti-apoptotic BCL-2 family proteins. Mechanistically, WSB assembles a CRL5 E3 ubiquitin ligase complex that facilitates the ubiquitination and subsequent proteasomal degradation of NOXA, a pro-apoptotic BCL-2 family protein. Loss of WSB2 leads to a substantial accumulation of NOXA in both cultured cell lines and knockout mouse tissues. While WSB2 deficiency alone does not significantly impact spontaneous apoptosis, it sensitizes cells to apoptosis when anti-apoptotic BCL-2 family proteins are either genetically depleted or pharmacologically inhibited. Moreover, WSB2 is overexpressed in several human cancer types. These findings identify WSB2 as a critical regulator of mitochondrial apoptosis and reveal the dysregulation of the WSB2–NOXA axis

as a key factor contributing to apoptosis resistance in cancer cells. Targeting both WSB2 and anti-apoptotic BCL-2 family proteins holds promising therapeutic potential for overcoming resistance in human cancers.

## Introduction

One of the defining hallmarks of human cancers is their ability to evade apoptosis, a form of programmed cell death essential for maintaining cellular homeostasis (*Hanahan and Weinberg, 2011*). This evasion not only drives tumor initiation and progression but also underlies resistance to many cancer therapies. Most anticancer treatments—including chemotherapy, radiotherapy, targeted therapy, and immunotherapy—rely on the activation of apoptotic pathways in cancer cells to achieve their therapeutic effects (*Carneiro and El-Deiry, 2020*). Apoptosis is predominantly regulated at the mitochondrial level by the B-cell lymphoma-2 (BCL-2) family of proteins, which are classified into anti-apoptotic and pro-apoptotic subgroups. Anti-apoptotic members such as BCL-2, BCL-XL, BCL-W, and MCL-1 contain one to four BH domains and preserve mitochondrial outer membrane integrity by neutralizing pro-apoptotic counterparts. In contrast, pro-apoptotic proteins include multidomain effectors (e.g., BAX, BAK, and BOK) and 'BH3-only' proteins (e.g., NOXA, BIM, PUMA, and BAD), which respond to cellular stress by triggering mitochondrial membrane permeabilization, leading to apoptosis (*Kale et al., 2018*; *Youle and Strasser, 2008*). Tumors frequently develop resistance to apoptosis by upregulating anti-apoptotic proteins or suppressing pro-apoptotic counterparts, emphasizing the need for therapeutic strategies that can restore this critical balance (*Cory et al., 2016*).

Small-molecule BH3 mimetics have emerged as a promising class of drugs that specifically inhibit anti-apoptotic BCL-2 proteins. Venetoclax (ABT-199), a BCL-2-specific inhibitor, has achieved clinical approval for treating BCL-2-dependent hematological malignancies, such as small lymphocytic lymphoma and chronic lymphocytic leukemia (*Cory et al., 2016*). Moreover, Venetoclax demonstrates enhanced efficacy in acute myeloid leukemia when combined with other therapies (*Wei et al., 2020*; *DiNardo et al., 2020*). However, its application in solid tumors remains limited, potentially due to the lower dependency of solid tumors on BCL-2 for survival (*Ploumaki et al., 2023*). Identifying alternative molecular targets to sensitize solid tumors to BCL-2 inhibitors represents a critical unmet need.

Cullin 5 (CUL5), a member of the cullin-RING ubiquitin ligase family, plays a pivotal role in protein turnover by assembling with RBX2, Elongin B/C, and a SOCS box-containing substrate-binding receptor to form the CRL5 E3 ubiquitin ligase complex (*Zhao et al., 2020*). Among these, WD repeat and SOCS box-containing protein 2 (WSB2) has been identified as a substrate receptor (*Mahrour et al., 2008*). WSB2 is frequently overexpressed in cancers such as lung, breast, and melanoma, where it promotes malignancy by driving cell proliferation, cycle progression, and migration (*Zhang et al., 2019*; *Ma et al., 2020*). Despite its oncogenic potential, the lack of identified physiological substrates for the CRL5$^{WSB2}$ complex has limited its exploration as a therapeutic target.

Using data from the DepMap Portal (https://depmap.org/), we uncovered a functional interplay between WSB2 and BCL-2 family proteins. Biochemical analyses revealed that WSB2 facilitates the degradation of NOXA, a pro-apoptotic BCL-2 family protein, through assembling the CRL5$^{WSB2}$ E3 ubiquitin ligase complex. Through a combination of cell line studies, xenograft tumor models, and knockout mice experiments, we investigated the biological significance and therapeutic potential of targeting WSB2. Our findings suggest that disrupting the WSB2–NOXA axis could synergize with BCL-2 inhibitors to induce apoptosis in cancer cells, offering a novel strategy for combating apoptosis resistance in human cancers.

## Results

### Multiple BCL-2 family proteins identified as interactors of WSB2

WSB2 has been identified as a substrate receptor of the CRL5 E3 ubiquitin ligase complex; however, its physiological roles in specific cellular processes remain largely unknown. Interestingly, several genome-wide RNAi and CRISPR/Cas9 screening studies have revealed strong synthetic interactions between WSB2 and key regulators of mitochondrial apoptosis, including MCL-1, BCL-xL, and MARCH5 (*McDonald et al., 2017*; *DeWeirdt et al., 2020*; *DeWeirdt et al., 2021*). To elucidate the potential molecular functions of WSB2, we analyzed the genetic co-dependency between WSB2 and

other proteins using Broad's 21Q2 DepMap dataset (*Tsherniak et al., 2017*). This dataset, derived from large-scale loss-of-function single-guide RNA (sgRNA) screens for vulnerabilities in 990 cancer cell lines, allows the identification of genes with similar functions or pathways (*Bayraktar et al., 2020*; *Price et al., 2019*). Gene ontology analysis of the top 100 WSB2 co-dependent genes revealed significant enrichment in apoptosis-related processes (*Figure 1—figure supplement 1A*, *Supplementary file 1* and *Supplementary file 2*). Among the most correlated genes, WSB2 displayed a positive association with anti-apoptotic proteins BCL2L2 (BCL-W) and MCL-1, while negatively correlating with pro-apoptotic proteins BAX and PMAIP1 (NOXA) (*Figure 1A*). Notably, WSB2, along with key BCL-2 family members (BCL-2, BCL-W, BAX, MCL-1, NOXA, and BAK1), constituted a co-essential module that also included UBE2J2/MARCH5, an E2–E3 ligase complex known to regulate MCL-1/NOXA turnover (*Figure 1B*; *Nakao et al., 2023*; *Djajawi et al., 2020*; *Haschka et al., 2020*).

Using the DepLink web server (*Nayak et al., 2023*) to analyze genetic and pharmacological perturbations (*Corsello et al., 2020*; *Iorio et al., 2016*), we observed that *WSB2* knockout exhibited the highest correlations with the BH3 mimetics ABT-737 and ABT-263 among hundreds of drugs tested (*Figure 1—figure supplement 1B, C*, *Supplementary file 3*). These molecular links prompted us to investigate whether WSB2 has any impact on mitochondrial apoptosis through the regulation of BCL-2 family proteins. To test this hypothesis, we first examined the interaction between WSB2 and BCL-2 family proteins. Exogenous co-immunoprecipitation (Co-IP) assays revealed that WSB2 interacted with MCL-1, NOXA, BAD, BCL-XL, and BCL-2 but not with BAX, BCL-W, or BAK (*Figure 1C*). These interactions were further confirmed through semi-endogenous and endogenous Co-IP assays (*Figure 1D, F–J*). In contrast, WSB1, a closely related paralog of WSB2, did not interact with BCL-2 family proteins (*Figure 1E*).

Since BCL-2 family proteins reside on the outer mitochondrial membrane, we sought to determine the subcellular distribution of WSB2. Immunofluorescence (IF) analysis demonstrated that WSB2 is predominantly cytoplasmic, with partial colocalization with the mitochondrial marker HSP60 (*Figure 1K*). Subcellular fractionation of HeLa cells further confirmed that WSB2 is primarily cytoplasmic, with a moderate mitochondrial presence and negligible nuclear localization (*Figure 1L*). To determine its submitochondrial localization, we purified mitochondria and performed Protease K digestion under different mitochondrial preparations. Cleavage by Protease K only occurred in intact mitochondria and targeted outer membrane proteins exposed to the cytosol, such as TOM70. Swelling the mitochondria with a hypotonic buffer disrupted the outer mitochondrial membrane but left the inner membrane intact, resulting in the cleavage of intermembrane space proteins like SMAC. Lysis with Triton X-100 caused the cleavage of all mitochondrial proteins including matrix proteins like HSP60. Similar to TOM70, WSB2 was cleaved by Protease K digestion in intact mitochondria preparation (*Figure 1M*). Cleavage patterns confirmed that WSB2 is exposed on the cytosolic side of the mitochondrial outer membrane.

Collectively, these data indicate that a proportion of WSB2 is located on the mitochondrial outer membrane, selectively interacting with certain members of the BCL-2 protein family.

## The CRL5^WSB2 E3 ubiquitin ligase complex mediates the ubiquitin–proteasomal degradation of NOXA

WSB2 was co-purified with CRL5 complex components (RBX2, CUL5, ELOB, and ELOC), confirming its role as a potential CRL5 adaptor (*Figure 1D*). To investigate whether WSB2 regulates the stability of BCL-2 family proteins, we analyzed protein levels following WSB2 overexpression or depletion. Overexpression of WSB2 markedly reduced NOXA levels, while the levels of other BCL-2 family proteins remained unaffected or minimally affected (*Figure 2—figure supplement 1A*). Furthermore, depletion of WSB2 through short hairpin RNA (shRNA)-mediated knockdown (KD) or CRISPR/Cas9-mediated knockout (KO) in prostate cancer C4-2B cells or liver cancer Huh-7 cells led to a marked increase in the steady-state levels of endogenous NOXA, without affecting other BCL-2 family proteins examined (*Figure 2A–C*, *Figure 2—figure supplement 2A, B*). Therefore, our main focus in this study was to investigate the impact of WSB2 on the protein stability of NOXA.

We further demonstrated that the proteasome inhibitor MG132 completely reversed the reduction in NOXA protein levels caused by WSB2 overexpression. In contrast, overexpression of WSB1 had no effect on NOXA levels (*Figure 2D*). WSB2 contains a C-terminal SOCS box, comprising a BC box and a Cullin 5 (CUL5) box, which are essential for interacting with Elongin B/C and CUL5, respectively

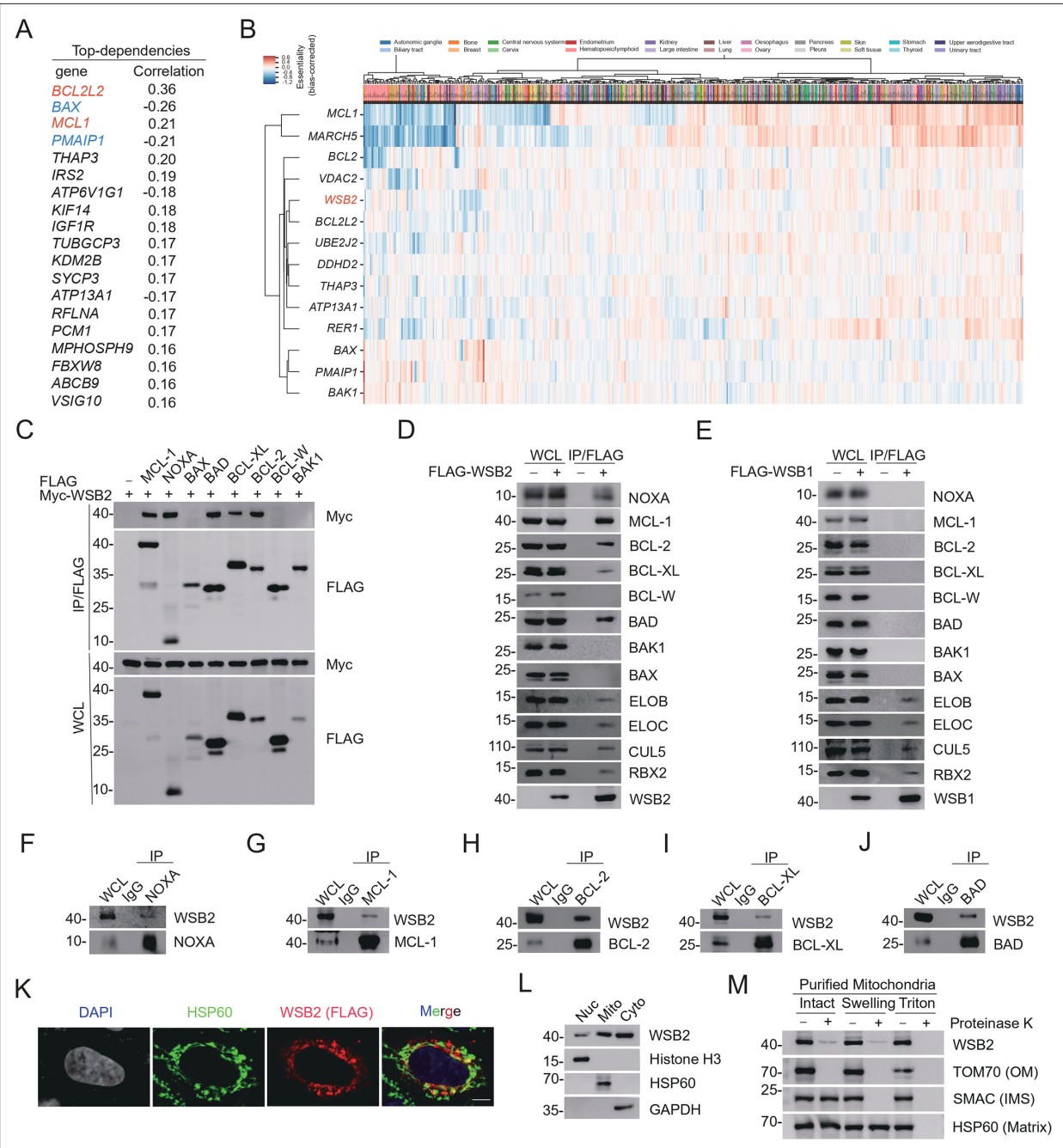

**Figure 1.** WSB2 interacts with multiple members of BCL-2 family proteins in cells. (**A**) The top 20 co-dependent genes of WSB2 in Broad's 21Q2 DepMap dataset. Red, anti-apoptotic BCL-2 family proteins. Blue, pro-apoptotic BCL-2 family proteins. (**B**) A co-essential module containing WSB2 and other BCL-2 family proteins using a dataset of CRISPR screens from the Achilles DepMap project (hhttps://mitra.stanford.edu/bassik/michael/cluster_heatmaps/). Cellular component: Bcl-2 family protein complex. Molecular function: BH domain binding. (**C–E**) Western blot (WB) analyses of the indicated proteins in the WCL and co-immunoprecipitation (Co-IP) samples of anti-FLAG antibody obtained from 293T cells transfected with the indicated plasmids. WB analyses of the indicated proteins in the WCL and Co-IP samples of IgG or anti-NOXA (**F**), anti-MCL-1 (**G**), anti-BCL-2 (**H**), anti-BCL-XL (**I**), or anti-BAD (**J**) antibodies obtained from 293T cells. (**K**) Representative immunofluorescence (IF) images from HeLa cells transfected with FLAG-WSB2, stained with FLAG (WSB2), HSP60, and DAPI. Scale bar, 20 μm. (**L**) The cytoplasmic (Cyto), mitochondrial (Mito), and nuclear fractions (Nuc) from HeLa cells were prepared as described in the Methods section. Histone H3 (nucleus), GAPDH (cytoplasm), and HSP60 (mitochondria) were used as subcellular fraction markers. WB analyses were performed to detect the indicated proteins in three fractions from HeLa cells. (**M**) Mitochondria from HeLa cells were purified as intact mitochondria or treated with hypotonic swelling buffer or lysed with Triton X-100 buffer. Different mitochondrial preparations were then digested with or without Proteinase K. WB analyses of the indicated proteins in three fractions were then performed. OM: outer membrane; IMS: intermembrane space.

*Figure 1 continued on next page*

*Figure 1 continued*

The online version of this article includes the following source data and figure supplement(s) for figure 1:

**Source data 1.** Original file for the western blot analysis in *Figure 1*.

**Source data 2.** Labeled file for the western blot analysis in *Figure 1*.

**Source data 3.** Original file for the images in *Figure 1*.

**Figure supplement 1.** WSB2 is involved in apoptosis-related pathways and shows significant associations with BH3 mimetic compounds.

---

(*Mahrour et al., 2008*). Deletion of either domain (BC box or CUL5 box) abolished WSB2's ability to decrease NOXA protein levels, as confirmed by experiments using WSB2 mutants (*Figure 2E*, *Figure 2—figure supplement 1B*). Reintroduction of these mutants into *WSB2* KO C4-2 cells failed to reverse the accumulation of NOXA, further confirming the functional importance of these domains (*Figure 2F*). Consistently, depletion of CRL5 complex components (RBX2, CUL5, ELOB, or ELOC) through siRNAs in C4-2B or Huh-7 cells also resulted in a significant increase in NOXA protein levels (*Figure 2G*, *Figure 2—figure supplement 1C*). Moreover, NOXA co-immunoprecipitated with all subunits of the CRL5$^{WSB2}$ complex (*Figure 2—figure supplement 1D*). Cycloheximide chase assays revealed that *WSB2* KO cells exhibited an extended half-life for NOXA (*Figure 2H, I*). Additionally, the mRNA levels of NOXA were even reduced in *WSB2* KO cells compared to control cells, likely to counteract the accumulation of NOXA protein (*Figure 2J*). This suggests that the accumulation of NOXA in *WSB2* KO cells is primarily due to impaired protein degradation. We also found that WSB2-WT, but not the BCM or CULM mutant could promote polyubiquitination of NOXA (*Figure 2K, L*). A previous study showed that UBE2F promotes the survival of lung cancer cells by activating CRL5 to degrade NOXA via the K11 Linkag (*Zhou et al., 2017*). By employing linkage-specific K11/K48/K63-Ub mutants, we observed that WSB2-mediated ubiquitination of NOXA likely involves all the tested ubiquitin linkage types (*Figure 2—figure supplement 1E*). Conversely, *WSB2* KO cells showed reduced levels of ubiquitinated NOXA (*Figure 2M*), further confirming that WSB2 is essential for NOXA turnover.

Collectively, these data indicate that the CRL5$^{WSB2}$ complex mediates the ubiquitin–proteasomal degradation of NOXA.

## The C-terminal region of NOXA is crucial for WSB2-mediated NOXA degradation

In addition to its BC and CUL5 boxes, WSB2 also contains five WD repeats, which typically function as modules facilitating protein–protein interactions. To identify the region responsible for binding NOXA, we generated a series of WSB2 deletion mutants and performed Co-IP assays. Surprisingly, we found that the SOCS box, rather than the WD repeats, is essential for WSB2 binding to NOXA (*Figure 3A, B*, *Figure 3—figure supplement 1A*). Deletion of the SOCS box abolished WSB2's ability to interact with NOXA and to mediate its ubiquitination and degradation, indicating that this domain is indispensable for its function (*Figure 3C, D*).

Reciprocally, we sought to determine the region in NOXA that is required for its interaction with WSB2. By generating a series of NOXA deletion mutants and conducting Co-IP assays, we identified the C-terminal region (40–54 aa), which contains the mitochondrial-targeting domain (MTD) (*Seo et al., 2003*), as the critical binding site (*Figure 3E–G*). Mutation of key residues within this domain (5A mutant) abolished the interaction between NOXA and WSB2, whereas mutations in the BH3 domain (3E mutant) did not affect their interaction (*Figure 3H*, *Figure 3—figure supplement 1B*). Furthermore, the NOXA-5A mutant was resistant to WSB2-mediated ubiquitination and degradation (*Figure 3I, J*), resulting in an extended protein half-life compared to NOXA-WT (*Figure 3K, L*). NOXA contains three lysine residues that can be attached by ubiquitin (*Pang et al., 2014*). By simultaneously mutating these lysine residues to arginine, we found that WSB2-mediated NOXA ubiquitination was completely abolished, although this mutant (KR) exhibits a comparable WSB2-binding capacity to NOXA-WT (*Figure 3H, J*). Additional in vivo ubiquitination assays revealed that lysine 48 is the primary residue mediating WSB2-dependent ubiquitination of NOXA (*Figure 3—figure supplement 1C*). These results indicate that the C-terminal region of NOXA, particularly its mitochondrial-targeting domain and lysine 48, is essential for its recognition and ubiquitination by WSB2.

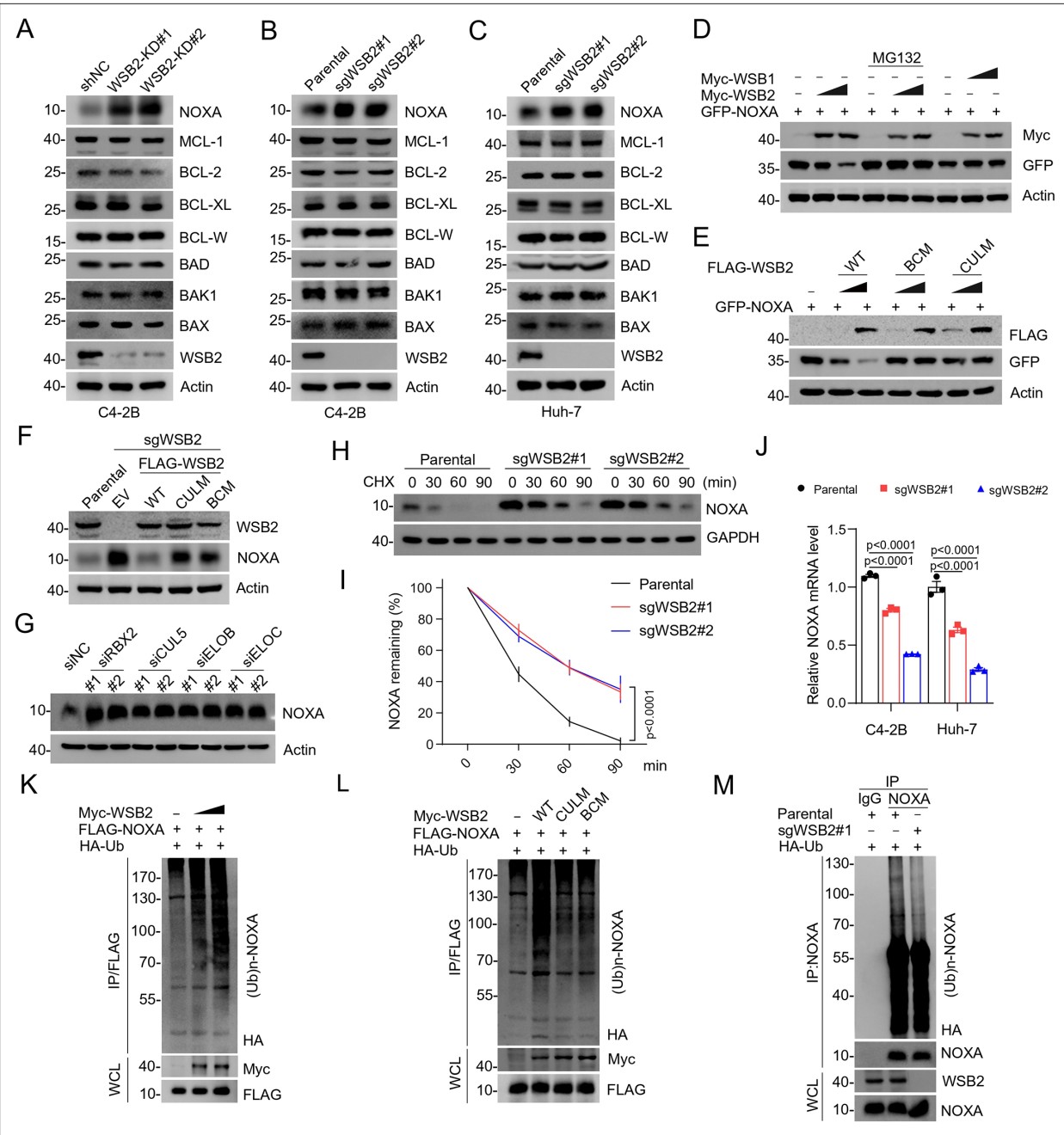

**Figure 2.** The CRL5$^{WSB2}$ E3 ubiquitin ligase complex controls NOXA protein turnover. (**A**) Western blot (WB) analyses of the indicated proteins in the WCL from C4-2B cells infected with lentivirus expressing WSB2-specific short hairpin RNA (shRNA) or negative control (NC). (**B, C**) WB analyses of the indicated proteins in the WCL from parental or WSB2 KO C4-2B or Huh-7 cells. (**D**) WB analyses of the indicated proteins in the WCL from 293T cells transfected with the indicated plasmids for 24 hr and treated with DMSO or MG132 (20 μM) for 8 hr. (**E**) WB analyses of the indicated proteins in the WCL from 293T cells transfected with the indicated plasmids. (**F**) WB analyses of the indicated proteins in the WCL from parental or WSB2 KO C4-2B cells stably overexpressing empty vector (EV), WSB2-WT or its mutants. (**G**) WB analyses of the indicated proteins in the WCL from C4-2B cells transfected with the indicated siRNAs. (**H, I**) WB analyses of indicated proteins in the WCL of parental and WSB2 KO C4-2B cells treated with cycloheximide (CHX, 50 μg/ml) and harvested at different time points. (**I**) At each time point, the intensity of NOXA was normalized to the intensity of GAPDH and then to the value at 0 hr. Data are shown as means ± SE ($n = 3$). (**J**) RT-qPCR measurement of NOXA mRNA expression in parental or WSB2 KO C4-2B or Huh-7 cells. Data are shown as means ± SE ($n = 3$). (**K, L**) WB analyses of the products of in vivo ubiquitination assays performed using WCL from 293T cells transfected with the indicated plasmids and treated with MG132 (20 μM). (**M**) WB analyses of the products of in vivo ubiquitination assays. Co-immunoprecipitation (Co-IP) using anti-IgG or anti-NOXA antibody in the WCL prepared from parental and *WSB2* KO C4-2B cells transfected with HA-Ub for 24 hr and treated with MG132 (20 μM) for 8 hr. p values are calculated by the two-way ANOVA test in (**I**) and one-way ANOVA test in (**J**). n.s., non-significant.

*Figure 2 continued on next page*

*Figure 2 continued*

The online version of this article includes the following source data and figure supplement(s) for figure 2:

**Source data 1.** Original file for the western blot analysis in *Figure 2*.

**Source data 2.** Labeled file for the western blot analysis in *Figure 2*.

**Figure supplement 1.** The CRL5$^{WSB2}$ E3 ubiquitin ligase complex controls NOXA protein turnover.

**Figure supplement 1—source data 1.** Original file for the western blot analysis in *Figure 2—figure supplement 1*.

**Figure supplement 1—source data 2.** Labeled file for the western blot analysis in *Figure 2—figure supplement 1*.

**Figure supplement 2.** Validation of *WSB2* knockout in two cancer cell lines.

To explore the therapeutic potential of disrupting the WSB2–NOXA interaction, we synthesized a peptide derived from the C-terminal region of NOXA (40–54 aa). This peptide competitively inhibited the interaction between WSB2 and NOXA in a dose-dependent manner (*Figure 3M*). Based on this, we hypothesized that transduction of the C-terminal NOXA peptide into cells could competitively inhibit WSB2-mediated NOXA degradation. To efficiently deliver this peptide into cells, we synthesized a fusion peptide in which the C-terminal peptide was connected to the cell-penetrating poly-arginine (R8) sequence. Treatment of cells with this fusion peptide reduced endogenous WSB2-NOXA interaction (*Figure 3—figure supplement 1D*), dose-dependently increased endogenous NOXA protein levels, and prolonged endogenous NOXA turnover (*Figure 3N*, *Figure 3—figure supplement 1E, F*).

Collectively, these data indicate that the C-terminal region of NOXA is indispensable for its interaction with WSB2 and subsequent ubiquitination.

## Co-inhibition of WSB2 and anti-apoptotic BCL-2 family proteins causes synthetic lethality via apoptotic cell death

Despite the significant accumulation of NOXA in WSB2-deficient cells, we did not observe obvious spontaneous apoptosis under standard cell culture conditions. This suggests that NOXA upregulation alone is insufficient to trigger spontaneous apoptosis.

Consequently, we hypothesized that simultaneous depletion of an anti-apoptotic BCL-2 family protein might synergize with WSB2 deficiency to induce apoptosis. To test this, we established stable cell lines with BCL-XL or MCL-1 KD via shRNAs, followed by further depletion of WSB2 using siRNA. Remarkably, co-depletion of BCL-XL and WSB2 or MCL-1 and WSB2 resulted in robust apoptosis, as evidenced by increased caspase cleavage in western blot (WB) analyses and a significant rise in apoptotic marker detected by flow cytometry (*Figure 4A–D*). The E3 ubiquitin ligase MARCH5 co-exists with WSB2 in a functional module (*Figure 1B*), and previous studies have shown that depletion of MARCH5 sensitizes cells to MCL-1 inhibitors or BCL-2/BCL-XL inhibitors (*Nakao et al., 2023*; *Subramanian et al., 2016*; *Arai et al., 2020*). Consistently, we observed that co-depletion of MARCH5 and WSB2 also induced substantial apoptosis (*Figure 4E, F*), supporting their cooperative role in regulating mitochondrial apoptosis.

Next, we used pharmacological inhibitors targeting anti-apoptotic BCL-2 family proteins to assess their efficacy in WSB2-deficient cells. In parental C4-2B and Huh-7 cells, treatment with either ABT-737 (an inhibitor of BCL-2/BCL-XL) or AZD5991 (an inhibitor of MCL-1) resulted in only modest apoptosis induction. However, in WSB2-deficient cells, these drugs caused substantial apoptosis (*Figure 4G–J*, *Figure 4—figure supplement 1A–D*). In addition to direct inhibitors of the BCL-2 family proteins, inhibitors of cyclin-dependent kinase 9 (CDK9) can indirectly target MCL-1 by suppressing the transcriptional activation of MCL-1 mRNAs (*Kabir et al., 2019*). Indeed, we found that the CDK9 inhibitor BAY-1143572 induced moderate apoptosis in parental cells but induced substantial apoptosis in WSB2-deficient cells (*Figure 4K, L*).

Collectively, these data indicate that co-targeting WSB2 and anti-apoptotic BCL-2 family proteins triggers synthetic lethality in cancer cells.

## The anti-apoptotic function of WSB2 is primarily reliant on NOXA downregulation

To determine whether WSB2's anti-apoptotic function is mediated primarily through its regulation of NOXA, we knocked down NOXA expression in WSB2-deficient C4-2B and Huh-7 cells using shRNA.

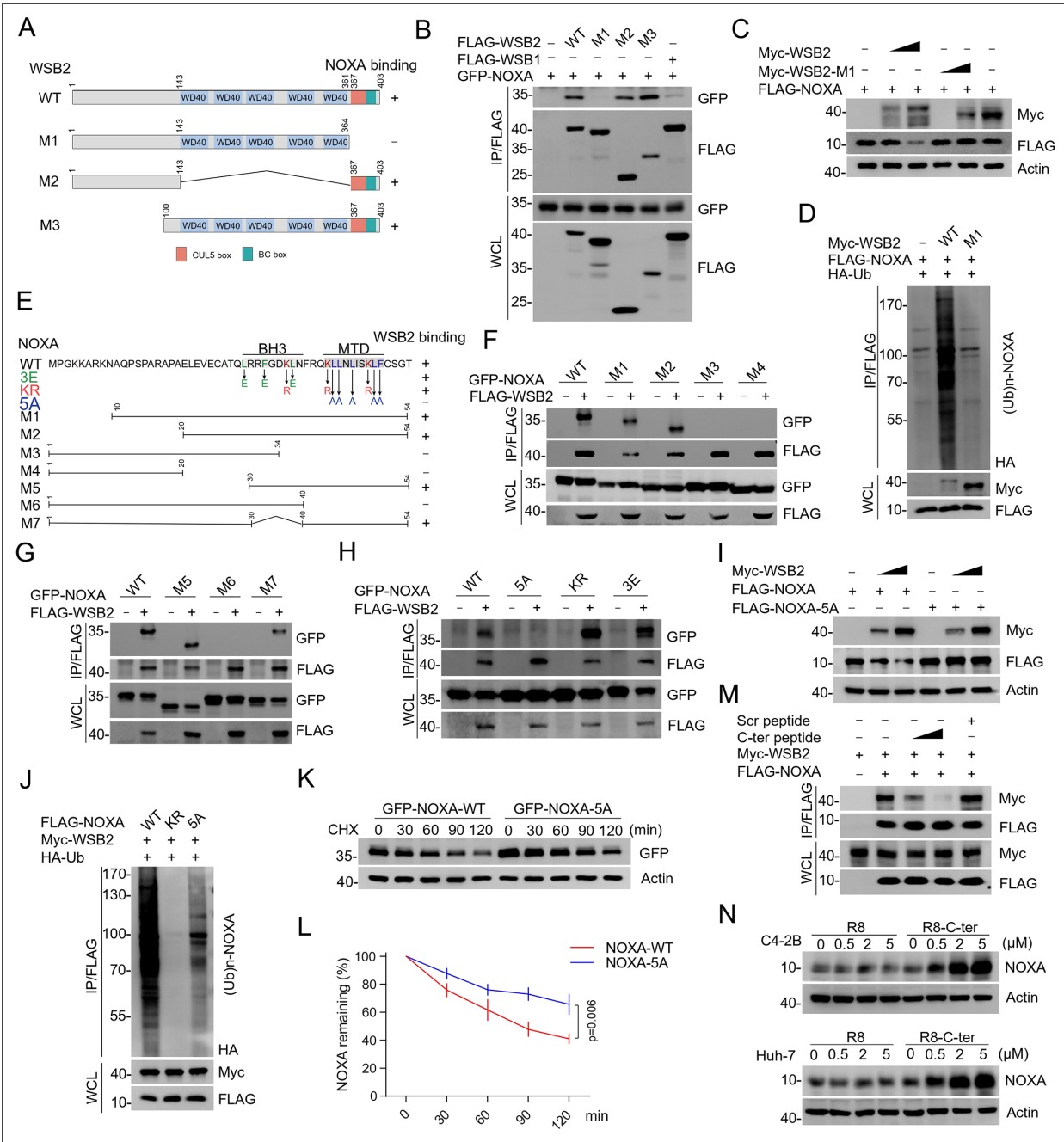

**Figure 3.** Identification of the mutual-binding regions of WSB2 and NOXA. (**A**) Schematic representation of WSB2 deletion mutants. (**B**) Western blot (WB) analyses of the indicated proteins in the WCL and co-immunoprecipitation (Co-IP) samples of anti-FLAG antibody obtained from 293T cells transfected with the indicated plasmids. (**C**) WB analyses of the indicated proteins in the WCL from 293T cells transfected with the indicated plasmids. (**D**) WB analyses of the products of in vivo ubiquitination assays performed using WCL from 293T cells transfected with the indicated plasmids for 24 hr and treated with MG132 (20 μM). (**E**) Schematic representation of NOXA deletion or point mutants. (**F–H**) WB analyses of the indicated proteins in the WCL and Co-IP samples of anti-FLAG antibody obtained from 293T cells transfected with indicated plasmids. (**I**) WB analyses of the indicated proteins in the WCL from 293T cells transfected with the indicated plasmids. (**J**) WB analyses of the products of in vivo ubiquitination assays performed using WCL from 293T cells transfected with the indicated plasmids and treated with MG132 (20 μM). (**K, L**) WB analyses of the indicated proteins in the WCL from 293T cells transfected with the indicated plasmids treated with cycloheximide (CHX, 50 μg/ml) and harvested at different time points. (**L**) At each time point, the intensity of NOXA was normalized to the intensity of GAPDH and then to the value at 0 hr. Data are shown as means ± SE (*n* = 3). p values are calculated by the two-way ANOVA test. (**M**) 293T cells were transfected with the indicated plasmids. WSB2–NOXA complex was co-immunoprecipitated by anti-FLAG antibody, and then the bound complex was incubated with the increasing amounts of C-terminal NOXA peptide (200 and 400 μg/ml) or the corresponding scramble peptide for 12 hr. Bound material was subjected to WB analyses. C-ter: C-terminal NOXA peptide. (**N**) WB analyses of the

*Figure 3 continued on next page*

*Figure 3 continued*

indicated proteins in the WCL from C4-2B and Huh-7 cells treated with the increasing concentration of C-terminal cell-penetrating peptide of NOXA or the R8 peptide for 12 hr.

The online version of this article includes the following source data and figure supplement(s) for figure 3:

Source data 1. Original file for the western blot analysis in *Figure 3*.

Source data 2. Labeled file for the western blot analysis in *Figure 3*.

Figure supplement 1. Identification of the mutual-binding regions of WSB2 and NOXA.

Figure supplement 1—source data 1. Original file for the western blot analysis in *Figure 3—figure supplement 1*.

Figure supplement 1—source data 2. Labeled file for the western blot analysis in *Figure 3—figure supplement 1*.

Strikingly, reducing NOXA levels largely reversed, though not completely, the substantial apoptosis induced by ABT-737 treatment (*Figure 5A, D*). Similar effects were observed when cells were treated with the MCL-1 inhibitor AZD5991, where NOXA KD substantially mitigated apoptosis in WSB2-deficient cells (*Figure 5E, F*). These results underscore the pivotal role of NOXA in mediating the apoptotic sensitivity of cells lacking WSB2.

We further explored whether NOXA upregulation could enhance sensitivity to BCL-2 inhibitors in cancer cells. Huh-7 cells treated with a cell-penetrating peptide derived from the C-terminal region of NOXA exhibited significantly increased sensitivity to ABT-737 (*Figure 5G, H*). This effect was dose dependent, with higher concentrations of the peptide correlating with greater accumulation of NOXA protein and higher levels of apoptosis. Importantly, the control peptide had no such effect. To validate these findings in vivo, we conducted xenograft tumor assays. Huh-7 tumors treated with either ABT-737 or the C-terminal NOXA peptide showed moderate reductions in tumor growth. Remarkably, co-administration of ABT-737 and the C-terminal NOXA peptide resulted in synergistic tumor inhibition, demonstrating the potential therapeutic efficacy of targeting the WSB2-NOXA axis in combination with BCL-2 family inhibitors (*Figure 5I, J*).

Collectively, these data indicate that WSB2 deficiency-induced hypersensitivity to BCL-2 family protein inhibitors was at least in part caused by NOXA accumulation.

## *Wsb2* knockout mice are more susceptible to apoptosis triggered by Venetoclax

To investigate the physiological role of WSB2 in apoptosis in vivo, we established a Wsb2 knockout mouse model (*Figure 6—figure supplement 1A*). Homozygous $Wsb2^{-/-}$ mice were observed to be born at Mendelian ratios and exhibited a normal lifespan with no apparent morphological or behavioral abnormalities (*Figure 6—figure supplement 1B, C*). Although some $Wsb2^{-/-}$ mice displayed reduced body size after birth, their adult size generally matched that of wild-type mice. At week 4, we collected multiple mouse tissues, including heart, liver, lung, kidney, and brain. WB analyses demonstrated varying degrees of upregulation in NOXA proteins in the tissues from $Wsb2^{-/-}$ mice compared to WT littermates. A strong upregulation of NOXA proteins was observed in the liver and heart tissues from $Wsb2^{-/-}$ mice, but not in lung, kidney, and brain tissues, indicating WSB2 modulates NOXA protein levels in a tissue-specific manner. However, the protein levels of caspases 3, 7, and 9 in these tissues were comparable between $Wsb2^{-/-}$ mice and WT littermates (*Figure 6—figure supplement 1D*). IF and WB analyses of heart and liver tissues revealed that cleaved caspases were nearly undetectable in both $Wsb2^{-/-}$ mice and WT littermates (*Figure 6A–E*). Consistent with the results from in vitro cell culture, Wsb2 ablation alone was insufficient to induce significant apoptosis in mouse organs.

To evaluate whether pharmacological inhibition of anti-apoptotic BCL-2 family proteins could trigger significant apoptosis in the organs from $Wsb2^{-/-}$ mice, we administered the BCL-2-specific inhibitor ABT-199 (Venetoclax) by oral gavage for 7 days. In the liver and heart tissues of ABT-199-treated $Wsb2^{-/-}$ mice, we detected pronounced apoptosis, as evidenced by increased levels of cleaved caspases, cleaved PARP1, and TUNEL-positive cells, which were absent in WT littermates under the same treatment conditions (*Figure 6A–E*, *Figure 6—figure supplement 2A–I*). These findings were corroborated by in vitro experiments, where primary hepatocytes isolated from $Wsb2^{-/-}$ mice exhibited significantly greater susceptibility to ABT-199-induced apoptosis compared to wild-type controls (*Figure 6F*).

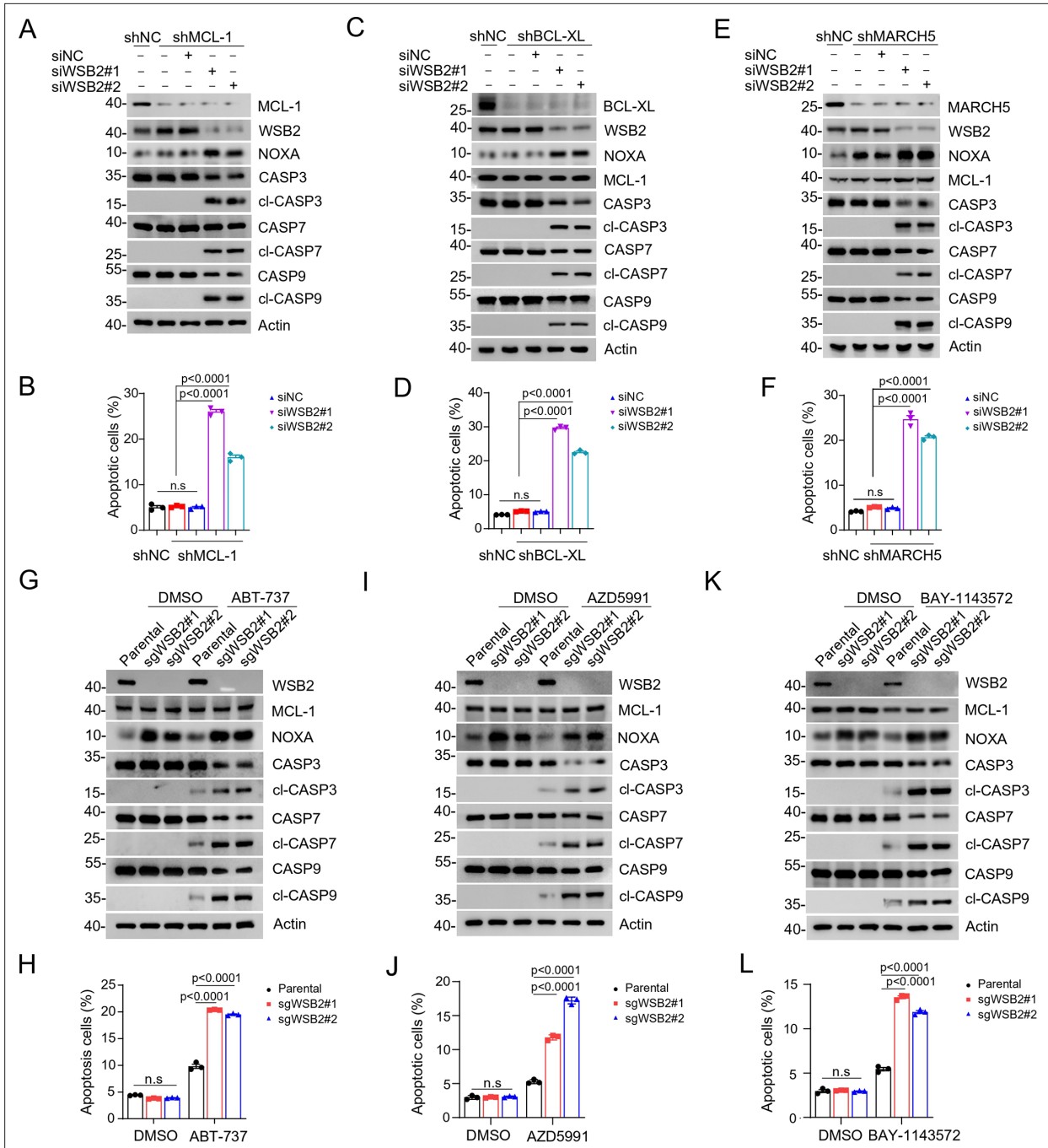

**Figure 4.** Combined inhibition of anti-apoptotic BCL-2 family members and WSB2 induces synthetic lethality. (**A, B**) Western blot (WB) analyses of the indicated proteins in the WCL from C4-2B cells infected with lentivirus expressing MCL-1-specific short hairpin RNA (shRNA) or NC and then transfected with the indicated siRNAs for 36 hr. (**B**) Annexin-V-FITC/PI assays were used to stain the harvested cells in (**A**), of which later flow cytometry analysis was performed. Data are shown as means ± SE (*n* = 3). (**C, D**) Similar to (**A, B**), but BCL-XL was knocked down in C4-2B cells. (**E, F**) Similar to (**A, B**), but MARCH5 was knocked down in C4-2B cells. (**G, H**) WB analyses of the indicated proteins in the WCL from parental and *WSB2* KO C4-2B cells treated with ABT-737 (20 μM) for 6 hr. (**H**) Annexin-V-FITC/PI assays were used to stain the harvested cells in (**H**), of which later flow cytometry analysis was performed. Data are shown as means ± SE (*n* = 3). (**I, J**) Similar to (**A, B**), but AZD5991 was used for treatment in C4-2B cells. (**K, L**) Similar to (**A, B**), but BAY-1143572 was used for treatment in C4-2B cells. p values are calculated by the one-way ANOVA test in (**B, D, F**) and two-way ANOVA test in (**H, J, L**). n.s., non-significant.

The online version of this article includes the following source data and figure supplement(s) for figure 4:

**Source data 1.** Original file for the western blot analysis in *Figure 4*.

*Figure 4 continued on next page*

*Figure 4 continued*

**Source data 2.** Labeled file for the western blot analysis in *Figure 4*.

**Figure supplement 1.** Combined inhibition of anti-apoptotic BCL-2 family members and WSB2 induces synthetic lethality.

**Figure supplement 1—source data 1.** Original file for the western blot analysis in *Figure 4—figure supplement 1*.

**Figure supplement 1—source data 2.** Labeled file for the western blot analysis in *Figure 4—figure supplement 1*.

To assess whether the cardiac injury was caused by ABT-199 treatment, we measured the levels of several cardiac enzyme markers, including CK (creatine kinase), CK-MB (creatine kinase isoenzyme MB), α-HBDH (α-hydroxybutyrate dehydrogenase), and LDH (lactate dehydrogenase) in serum. As shown in *Figure 6G*, ABT-199 administration led to a significant elevation in the levels of these cardiac enzymes in *Wsb2⁻/⁻* mice, whereas no such effect was observed in WT littermates. These results suggest that co-inhibition of WSB2 and BCL-2 exacerbates apoptosis in cardiomyocytes, leading to tissue injury. To investigate whether the anti-apoptotic function of WSB2 is primarily reliant on mouse NOXA, we isolated mouse embryonic fibroblasts (MEFs). NOXA protein levels were markedly upregulated in *Wsb2⁻/⁻* MEFs compared to wild-type controls (*Figure 6H*). Moreover, reducing the expression of NOXA through shRNA-mediated KD in *Wsb2⁻/⁻* MEFs largely reversed the substantial apoptosis induced by ABT-199 treatment (*Figure 6I*).

Collectively, these data indicate that WSB2-mediated NOXA destabilization is evolutionarily conserved, and this regulatory axis is critical for maintaining tissue homeostasis.

## WSB2 is overexpressed in several human cancer types

To explore the clinical relevance of WSB2, we analyzed its expression across various human cancer types using RNA-sequencing (RNA-seq) data from The Cancer Genome Atlas (TCGA). Remarkably, WSB2 mRNA levels were significantly elevated in multiple cancers, including prostate adenocarcinoma (PRAD) and liver hepatocellular carcinoma (LIHC), compared to corresponding normal tissues (*Figure 7A*). In PRAD, higher WSB2 expression positively correlated with key clinical parameters such as Gleason score (*Figure 7B*), pathological stage (*Figure 7C*), clinical stage (*Figure 7D*), and nodal metastasis status (*Figure 7E*). Similarly, in LIHC, increased WSB2 expression was associated with higher clinical stage (*Figure 7F*), pathological grade (*Figure 7G*), and nodal metastasis status (*Figure 7H*). Kaplan–Meier survival analysis revealed that high WSB2 expression was significantly linked to shorter overall survival in LIHC patients, while no such association was observed in PRAD (*Figure 7I, J*).

To validate these findings, we conducted immunohistochemistry (IHC) analysis using a WSB2-specific antibody after confirming its specificity (*Figure 7—figure supplement 1A*). Notably, IHC analysis of a PRAD tissue microarray revealed a positive correlation between WSB2 protein expression and Gleason score (*Figure 7K, L*). IHC analysis of an LIHC tissue microarray revealed a positive correlation between WSB2 protein expression and clinical grade (*Figure 7M, N*). Moreover, analysis of publicly available RNA-seq datasets revealed that WSB2 expression was significantly higher in sorafenib-resistant LIHC patients compared to sorafenib-sensitive patients, further underscoring its role in therapeutic resistance (*Figure 7O, P*).

Collectively, these data suggest that WSB2 overexpression is a common feature in several aggressive cancers and may contribute to tumor progression, metastasis, and resistance to therapy.

## Discussion

The WSB2 protein has been identified in several large-scale proteome mapping analyses as being co-purified from the CUL5 scaffold complex (*Huttlin et al., 2017*; *Bennett et al., 2010*). With the presence of a SOCS box in its protein sequence, WSB2 is believed to serve as a receptor for substrates, assisting in their recognition by the CRL5 E3 ubiquitin ligase complex. Its close paralog, WSB1, is induced by hypoxia and can form a CRL5^WSB1 complex that promotes cancer metastasis by inducing VHL degradation (*Kim et al., 2015*). Additionally, the CRL5^WSB1 complex overcomes oncogene-induced senescence by targeting ATM for degradation (*Kim et al., 2017*). However, the physiological substrates of the CRL5^WSB2 complex remain poorly understood. Nevertheless, prior to our current study, there are several pieces of evidence suggesting that this complex may play a role in regulating cell death, possibly involving BCL-2 family proteins. In a large-scale RNAi screening aimed

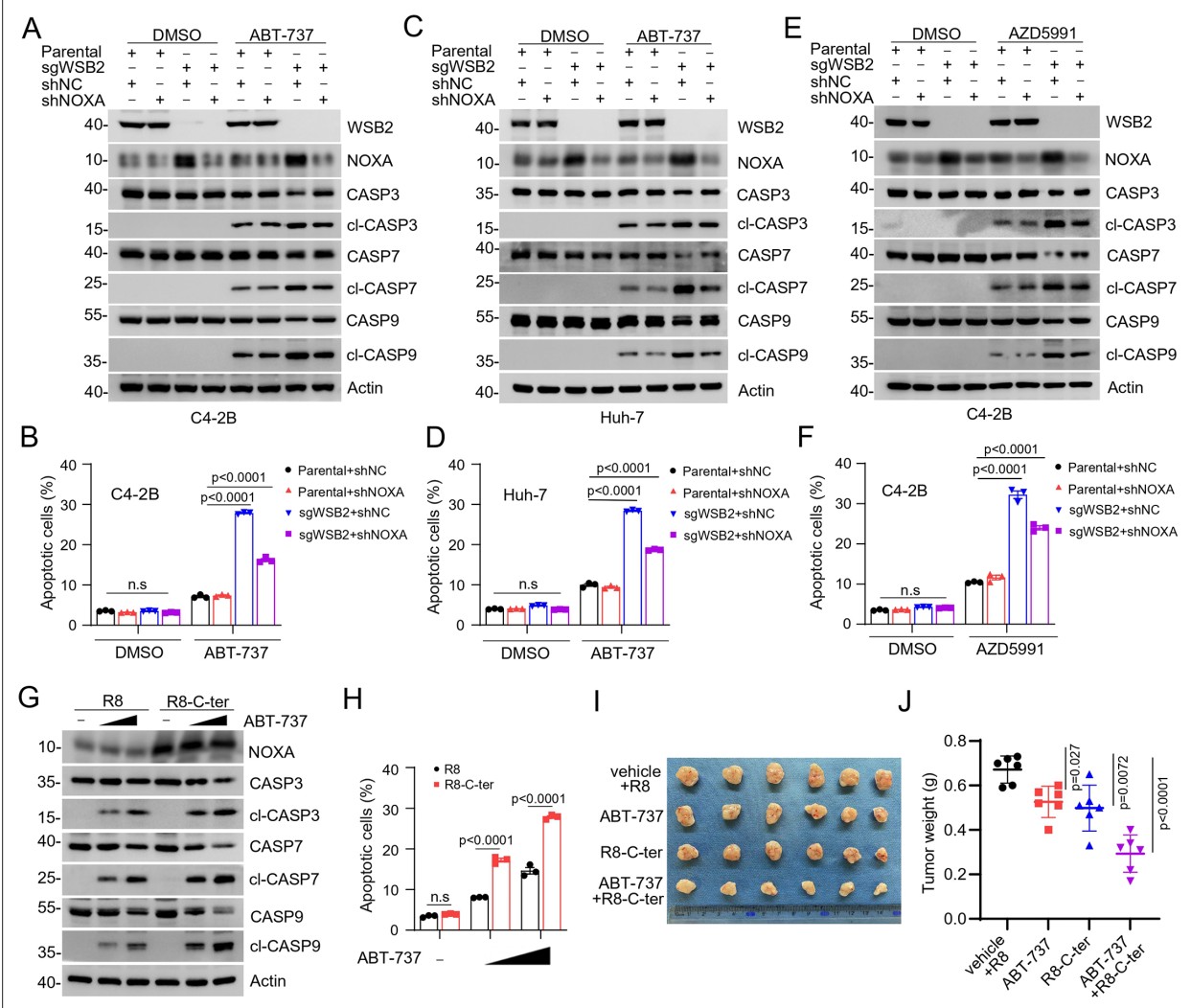

**Figure 5.** WSB2 primarily exerts its anti-apoptotic function largely via destabilizing NOXA. (**A, B**) Western blot (WB) analyses of indicated proteins in the WCL from parental or WSB2 KO C4-2B cells infected with lentivirus expressing NOXA-specific short hairpin RNA (shRNA) or NC, treated with DMSO or ABT-737 (20 μM) for 6 hr. (**B**) Annexin-V-FITC/PI assays were used to stain the harvested cells, of which later flow cytometry analysis was performed. Data are shown as means ± SE (*n* = 3). (**C, D**) Similar to (**A, B**), but AZD5991 (10 μM) was used for treatment in C4-2B cells. (**E, F**) Similar to (**A, B**), but Huh-7 cells were treated. (**G, H**) WB analyses of the indicated proteins in the WCL from Huh-7 cells treated with the cell-penetrating C-terminal NOXA peptide (5 μM) or the corresponding R8 peptide (5 μM) for 12 hr and then the cells were treated with increasing doses of ABT-737 (10 and 20 μM). Annexin-V-FITC/PI assays were used to stain the harvested cells, of which later flow cytometry analysis was performed. Data are shown as means ± SE (*n* = 3). (**I, J**) Huh-7 cells were injected subcutaneously (s.c.) into the right flank of BALB/c mice and treated with ABT-737 (30 mg/kg), R8-C-ter (20 mg/kg), or R8 (20 mg/kg) as control at different day points. 6 mice per experimental group. Tumors in each group at day 20 were harvested and photographed (**I**) and tumor weight (**J**) was documented. Data are shown as means ± SD (*n* = 6). p values are calculated by the two-way ANOVA test in (**B, D, F, H**). n.s., non-significant.

The online version of this article includes the following source data for figure 5:

**Source data 1.** Original file for the western blot analysis in *Figure 5*.

**Source data 2.** Labeled file for the western blot analysis in *Figure 5*.

at understanding cancer dependencies and synthetic lethal relationships, the top correlates of WSB2 co-essentiality were found to be MCL-1, BCL-2, and MARCH5, while the most strongly anti-correlated gene with WSB2 was BAX (*McDonald et al., 2017*). Two CRISPR/Cas9 knockout screens also showed strong synthetic relationships between WSB2 and MCL-1, BCL-2, or MARCH5 (*DeWeirdt et al., 2020*; *DeWeirdt et al., 2021*). A comprehensive phenotypic CRISPR/Cas9 screen of the ubiquitin pathway revealed that knockout of CUL5, RBX2, or WSB2 resulted in cells becoming hypersensitive to the CRM1 inhibitor leptomycin (*Hundley et al., 2021*). Furthermore, a genome-wide CRISPR inhibition

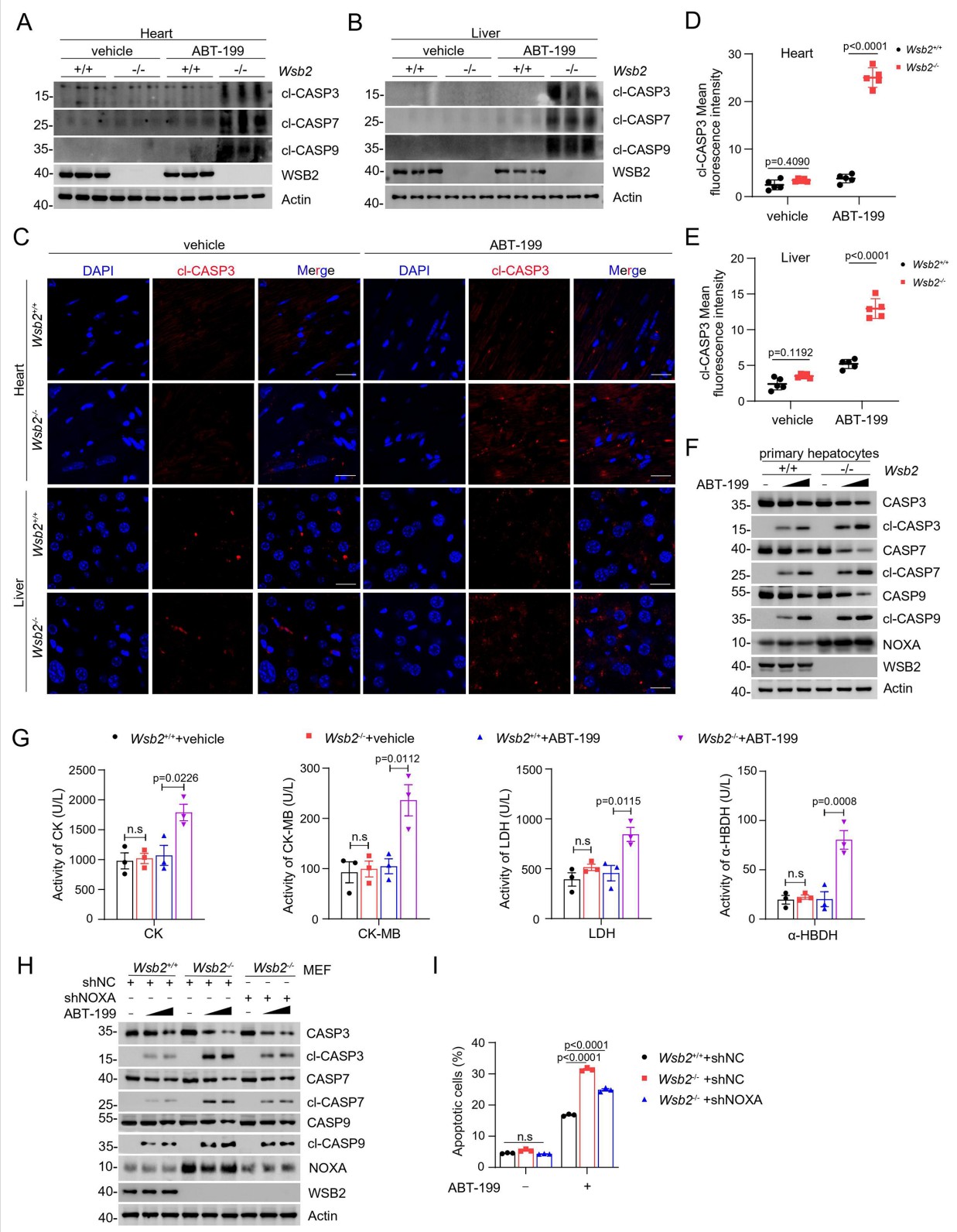

**Figure 6.** Validation of the anti-apoptotic function of WSB2 using Wsb2 knockout mouse models. Western blot (WB) analyses of the indicated proteins in the WCL from heart (**A**) or liver (**B**) tissues obtained from *Wsb2*[+/+] and *Wsb2*[−/−] mice after gavage administration of vehicle or ABT-199 (100 mg/kg/day) for 7 days. (**C**) Representative immunofluorescence (IF) images from the heart or liver tissues of *Wsb2*[+/+] and *Wsb2*[−/−] mice after oral administration of vehicle or ABT-199 (100 mg/kg/day) for 7 days, and stained with cl-CASP3 and DAPI. Scale bar, 20 μm. The mean fluorescence intensity of cl-

*Figure 6 continued on next page*

*Figure 6 continued*

CASP3 from the heart (**D**) or liver tissues (**E**) obtained from *Wsb2*[+/+] and *Wsb2*[−/−] mice. Data were shown as means ± SE (*n* = 5). (**F**) WB analyses of the indicated proteins in the WCL from the primary hepatocytes of *Wsb2*[+/+] and *Wsb2*[−/−] mice treated with ABT-199 (10 and 20 µM) for 6 hr. (**G**) The levels of myocardial zymogram in serum from *Wsb2*[+/+] and *Wsb2*[−/−] mice after gavage administration of vehicle or ABT-199 (100 mg/kg/day) for 7 days. (**H**) WB analyses of the indicated proteins in the WCL from *Wsb2*[+/+] and *Wsb2*[−/−] mouse embryonic fibroblasts (MEFs) infected with lentivirus expressing NOXA-specific short hairpin RNA (shRNA) or NC, treated with DMSO or ABT-199 (10 and 20 µM) for 6 hr. (**I**) Annexin-V-FITC/PI assays were used to stain the harvested cells, of which later flow cytometry analysis was performed. Data are shown as means ± SE (*n* = 3). p values are calculated by the one-way ANOVA test in (**G**) and two-way ANOVA test in (**D, E**). n.s., non-significant.

The online version of this article includes the following source data and figure supplement(s) for figure 6:

**Source data 1.** Original file for the western blot analysis in *Figure 6*.

**Source data 2.** Labeled file for the western blot analysis in *Figure 6*.

**Source data 3.** Original file for the images in *Figure 6*.

**Figure supplement 1.** Generation and validation of *Wsb2* KO mouse models.

**Figure supplement 1—source data 1.** Original file for the western blot analysis in *Figure 6—figure supplement 1*.

**Figure supplement 1—source data 2.** Labeled file for the western blot analysis in *Figure 6—figure supplement 1*.

**Figure supplement 2.** WSB2 deficiency enhances the susceptibility of ABT 199-induced apoptosis in the heart and liver tissues of *Wsb2*[−/−] mice.

**Figure supplement 2—source data 1.** Original file for the images in *Figure 6—figure supplement 2*.

(CRISPRi) screen conducted in lung cancer cells demonstrated that knockout of CUL5, RBX2, or UBE2F (a specific E2 for CRL5 E3s) caused cells to become hypersensitive to a CDK9 inhibitor or MCL-1 inhibitor (*Kabir et al., 2019*). By connecting the dots from these studies, our findings provide a comprehensive scenario. We have shown that WSB2 assembles an active CRL5[WSB2] complex, which mediates the ubiquitination and proteasomal turnover of NOXA, maintaining its low-level expression under basal conditions. WSB2 deficiency leads to a remarkable accumulation of NOXA, but it alone is not sufficient to trigger spontaneous apoptosis. This is consistent with previous studies showing that enforced expression of NOXA alone is ineffective at triggering apoptosis in various cell types (*Akhtar et al., 2006*; *Shibue et al., 2006*). Striking when combined with genetic or pharmacological inhibition of anti-apoptotic BCL-2 family proteins, massive apoptosis occurs in WSB2-deficient cells (*Figure 8*). However, it is important to note that KD of NOXA expression in WSB2-deficient cells largely, but not completely, reverses the massive apoptosis induced by BCL-2 family protein inhibitors, implying that WSB2 may also modulate apoptosis through other unidentified targets. Indeed, WSB2 interacts with multiple BCL-2 family proteins (MCL-1, BCL-2, BCL-XL, and BAD) (*Figure 1F–J*). Although WSB2 does not alter their turnover, it is still possible that WSB2 modulates the apoptotic function of these proteins through direct binding. Further investigation is warranted to fully elucidate the molecular mechanisms underlying WSB2-mediated anti-apoptotic function. Lastly, it would also be interesting to explore whether there are any upstream signals capable of overriding WSB2-mediated NOXA destabilization under specific stress conditions.

In the current study, our focus was primarily on investigating the in vivo anti-apoptotic function of WSB2. Thus, we did not extensively characterize the potential morphological and behavioral abnormalities in *Wsb2*[−/−] mice. However, recent large-scale mouse phenotype analyses conducted by the International Mouse Phenotyping Consortium (IMPC) have reported abnormalities in tooth morphology, locomotor activity, retina, heart, osmotic and electrolyte balance, as well as male infertility in Wsb2[−/−] mice (*da Silva-Buttkus et al., 2023*). It remains unclear whether these abnormalities are a result of NOXA accumulation and subsequent dysregulated apoptosis. Generating Wsb2/NOXA double knockout mice would be beneficial in determining whether these abnormalities can be reversed by further ablating NOXA expression. Despite the fact that WSB2 is upregulated in various types of cancer and exhibits a strong anti-apoptotic function, which makes it a promising target for cancer therapy, it is crucial to comprehensively understand the downstream substrates regulated by the CRL5[WSB2] E3 ligase complex. This insight will help us better evaluate the potential consequences of pharmacologically inhibiting WSB2, considering that such inhibition could disrupt the regulation of additional substrates, potentially resulting in unwanted side effects.

Since its initial discovery as a novel phorbol-12-myristate13-acetate responsive gene in T cells, and subsequently as a transcriptional target of the genotoxic response regulator p53, NOXA has emerged as a critical player in regulating cell death pathways in various cell types under stressed conditions

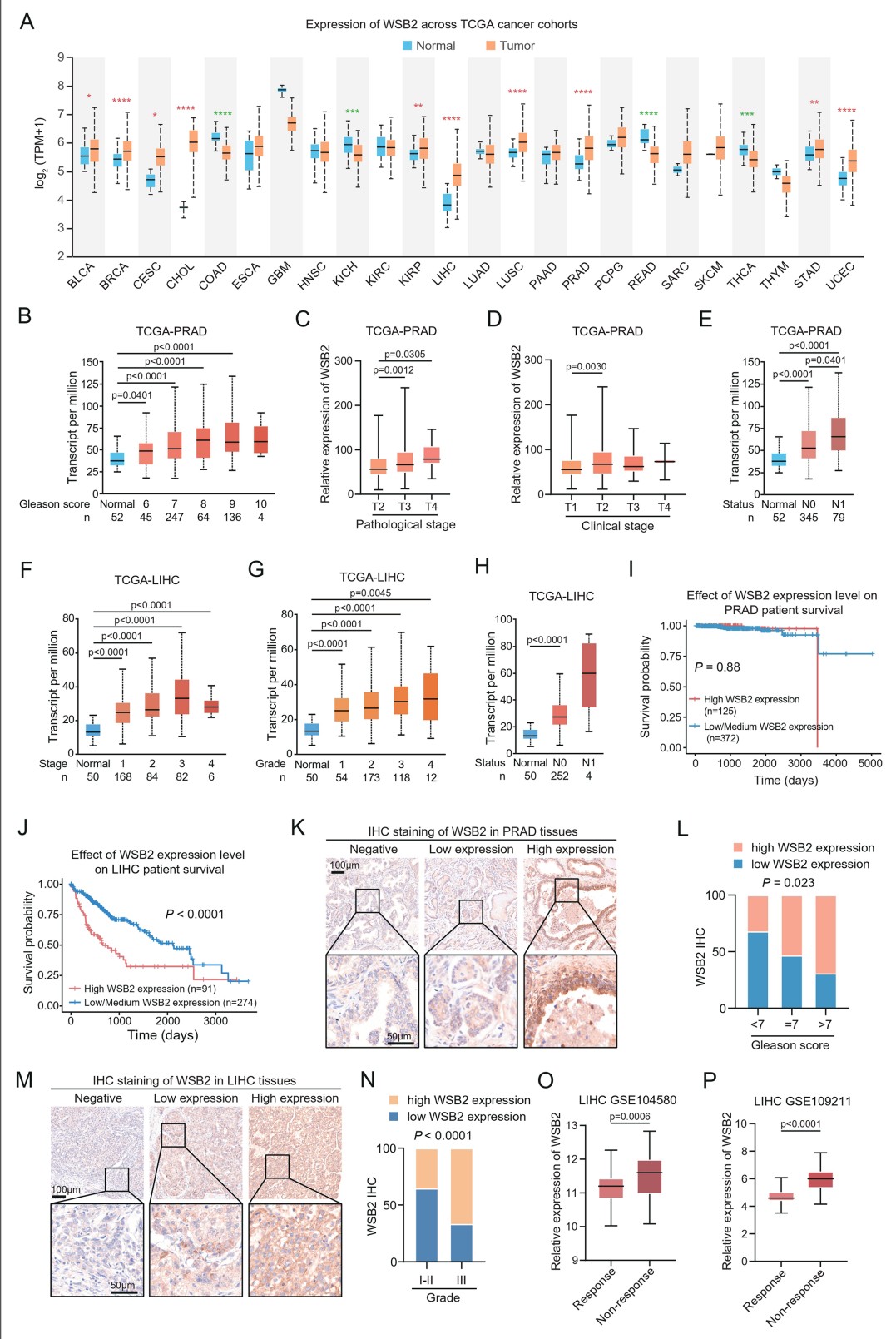

**Figure 7.** WSB2 expression is upregulated in multiple human cancers. (**A**) WSB2 mRNA expression in normal and tumor tissues from the The Cancer Genome Atlas (TCGA) cohort. Relationship between WSB2 mRNA expression and Gleason score (**B**), pathological T stage (**C**), clinical T stage (**D**), and nodal metastasis status (**E**) in prostate adenocarcinoma (PRAD) patients from the TCGA cohort. Relationship between WSB2 mRNA expression and

*Figure 7 continued on next page*

*Figure 7 continued*

clinical stage (**F**), pathological grade (**G**), and nodal metastasis status (**H**) in liver hepatocellular carcinoma (LIHC) patients from the TCGA cohort. Kaplan–Meier survival plots of overall survival (OS) according to WSB2 mRNA expression in PRAD (**I**) and LIHC (**J**) patients from the TCGA cohorts. (**K**) Representative immunohistochemistry (IHC) staining results for WSB2 in PRAD TMA, scale bar, 50 µm. (**L**) Quantification analysis of WSB2 IHC staining in PRAD patients by Gleason score categories. *n* = 84. (**M**) Representative IHC staining results for WSB2 in LIHC TMA, scale bar, 50 µm. (**N**) Quantification analysis of WSB2 IHC staining in LIHC patients by tumor grade categories. *n* = 29. (**O**) The WSB2 mRNA levels in 81 sorafenib-response and 66 sorafenib-non-response LIHC patients from GSE104580 dataset. (**P**) The WSB2 mRNA levels in 42 sorafenib-response and 98 sorafenib-non-response LIHC patients from GSE109211 dataset. p values are calculated by the unpaired *t* test in (**A–H, L, N–P**) and log-rank test in (**I, J**). *p<0.05, **p<0.01, ***p<0.001, ****p<0.0001.

The online version of this article includes the following source data and figure supplement(s) for figure 7:

**Source data 1.** Original file for the images in *Figure 7*.

**Figure supplement 1.** Validation of WSB2 antibody specificity by immunohistochemistry (IHC).

(*Ploner et al., 2008*). Notably, NOXA is implicated in fine-tuning apoptosis induction in cancer cells treated with genotoxic anticancer drugs, including paclitaxel (a microtubule targeting agent), bortezomib (a proteasome inhibitor), and MLN4924 (a CRL E3 ligase inhibitor) (*Ploner et al., 2008*). These different agents engage distinct mechanisms, such as transcriptional activation or protein stabilization, to upregulate NOXA protein levels and initiate apoptotic cell death (*Ploner et al., 2008*). The half-life

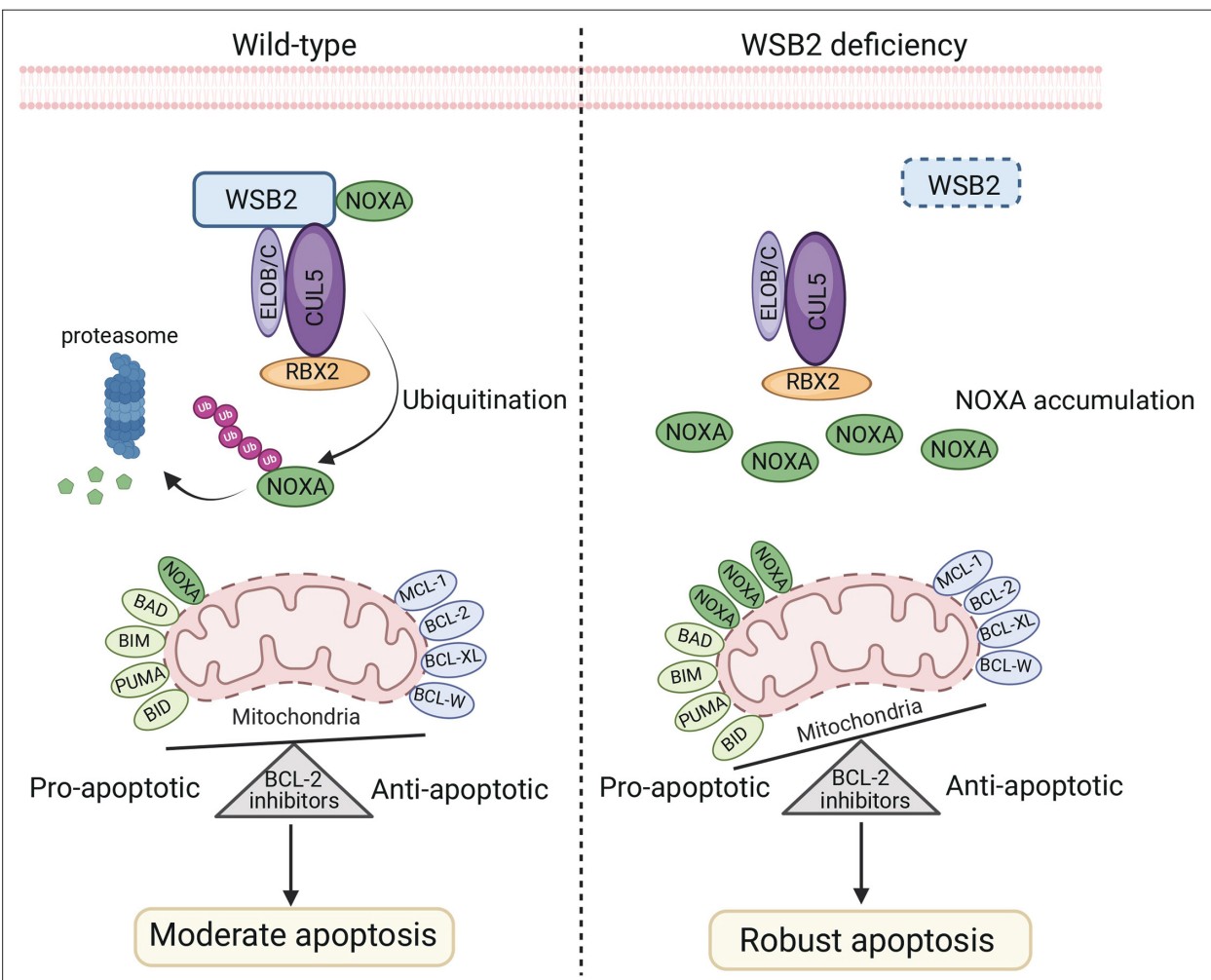

**Figure 8.** The schematic diagram illustrates that the disruption of the CRL5^WSB2 E3 ubiquitin ligase complex results in NOXA stabilization and increased sensitivity to BCL-2 family protein inhibitors. Created using BioRender.com.

of NOXA protein was very short, as it undergoes ubiquitin-proteasomal degradation mediated by the addition of ubiquitin to specific lysine residues (*Mahrour et al., 2008*). In mantle cell lymphoma (MCL) cell lines, despite high NOXA transcript levels, low NOXA protein expression is observed due to rapid protein degradation (*Dengler et al., 2014*). Similarly, paradoxical downregulation of NOXA protein is observed in Cushing's disease (CD) adenomas, despite transcriptional upregulation caused by recurrent promoter hypomethylation (*Asuzu et al., 2022*). These observations suggest that certain tumor cells may exploit pathways to accelerate NOXA degradation, thus suppressing apoptosis. The elevated NOXA mRNA levels observed in tumor cells may potentially serve as a compensatory mechanism to counteract the reduction in NOXA protein levels. Previous studies have only partially characterized the ubiquitin-proteasomal degradation of NOXA, showing that UBE2F, in conjunction with RBX2, induces CUL5 neddylation, leading to CRL5 E3 activation and subsequent NOXA degradation (*Kabir et al., 2019*). In lung cancer tissues, high levels of UBE2F and CUL5 correlate with reduced NOXA levels and poorer survival in patients. However, the specific CRL5 substrate receptor responsible for NOXA destabilization has yet to be identified (*Zhou et al., 2017*). In our study, we identified WSB2 as the substrate receptor for NOXA, thereby shedding light on its role in regulating NOXA turnover. Further investigation is needed to determine whether WSB2 dysregulation is responsible for the accelerated protein turnover observed in various cancer types, such as MCL and CD adenomas. It should also be noted that WSB2 only facilitates NOXA destabilization in certain tissues/organs, such as heart and liver, in mouse models. It is not surprising that a specific substrate can be targeted by multiple E3 ubiquitin ligases. A previous study has indicated that treatment with a proteasome inhibitor could further increase NOXA protein levels in CUL5 knockout cells, suggesting that the turnover of NOXA can be regulated by additional ubiquitin ligases apart from the CRL5 E3s (*Kabir et al., 2019*). In fact, the RING domain-containing E3 ubiquitin ligases MARCH5 have been reported to mediate NOXA degradation (*Nakao et al., 2023*; *Djajawi et al., 2020*; *Haschka et al., 2020*; *Subramanian et al., 2016*; *Arai et al., 2020*). Further investigation is needed to determine which E3 ubiquitin ligase(s) play the predominant roles in specific tissues/organs or types of cancer.

Although bortezomib and MLN4924 have proven effective in stabilizing the NOXA protein and promoting NOXA-dependent apoptosis, their broad inhibition of the proteasome or all CRL E3 ligases, respectively, inevitably leads to side effects. In this study, we have conducted preliminary investigations on the use of a competitive peptide to effectively inhibit the binding of WSB2 and NOXA, resulting in the accumulation of NOXA proteins and increased sensitivity to ABT-737. In the future, other potential therapeutic strategies can be explored, such as designing PROTAC molecules specifically for degrading WSB2 or developing small molecules to disrupt the interaction between WSB2 and NOXA. Extensive research is needed to determine the safety and efficacy of these approaches in preclinical cancer models.

## Materials and methods
### Acquisition and analysis of DepMap and drug sensitivity datasets
Gene co-dependencies were determined using the Achilles datasets (https://depmap.org/portal/). The Achilles dataset contains dependency scores from genome-scale essentiality screens scores of 789 cell lines. As a measure of co-dependency, the Pearson's correlation coefficient of essentiality scores was computed for all gene pairs. GO analysis for the top 500 genes co-dependent with WSB2 was performed using PANTHER to search enriched biological processes and pathways. Co-essential module assignments of cellular component: Bcl-2 family protein complex was obtained from a previously published dataset (Gene co-dependency, https://mitra.stanford.edu/bassik/michael/cluster_heatmaps/; *Wainberg et al., 2021*). To identify genetic and pharmacologic perturbations that induce similar effects on cell viability, a Web tool, DepLink (https://shiny.crc.pitt.edu/deplink/), was used (*Nayak et al., 2023*).

### Cell line, cell culture, transfection, and lentiviral infection
293T, HeLa, C4-2B, and Huh-7 cells were obtained from the American Type Culture Collection (ATCC). 293T, HeLa, and Huh-7 cells were maintained in Dulbecco's modified Eagle's medium (DMEM) supplemented with 10% fetal bovine serum (FBS). C4-2B cells were maintained in RPMI1640 medium supplemented with 10% FBS. We routinely perform DNA fingerprinting and PCR to verify the authenticity

of the cell lines and to ensure they are free of mycoplasma infection. We conducted transient transfection using EZ Trans (Shanghai Life-iLab Biotech). For lentiviral transfection, we transfected pLKO shRNA KD and virus-packing constructs into 293T cells. The viral supernatant was collected after 48 hr. The cells were then infected with the viral supernatant in the presence of polybrene (8 µg/ml) and selected in growth media containing puromycin (1.5 µg/ml). The gene-specific shRNA or siRNA sequences can be found in *Supplementary file 4*.

### Antibodies, chemicals, and kits

The information on antibodies, chemicals, and kits used in this study is listed in Appendix 1—key resources table.

### Plasmid construction

The plasmids used for transient overexpression were constructed using the pCMV-FLAG/Myc vector (Clontech). Point and deletion mutants were engineered utilizing the KOD-Plus-Mutagenesis Kit (TOYOBO) following the manufacturer's instructions. sgRNAs targeting WSB2 (https://crispor.gi.ucsc.edu) were subcloned into the pSpCas9(BB)-2A-Puro (PX459) vector for gene knockout (KO). shRNAs targeting WSB2 or BCL-2 family proteins were subcloned into the pLKO.1 puro vector (Addgene) for gene KD. The sequences of gene-specific sgRNAs and shRNAs are listed in *Supplementary file 4*.

### Isolation of nucleic, cytoplasmic, and mitochondrial fractions

HeLa cells were prepared for nuclear, cytoplasmic, and mitochondrial extraction by density-gradient centrifugation. Briefly, HeLa cells were washed three times with PBS. Then the cells are suspended by using hypotonic solution (140 mM KCl, 10 mM EDTA, 5 mM $MgCl_2$, 20 mM HEPES (pH 7.4), and the protease inhibitor). Then $5 \times 10^6$ HeLa cells were ground with a glass homogenizer in an ice bath for 25 strokes. Nuclear, cytoplasmic, and mitochondrial fractions were separated through differential centrifugation ($800 \times g$, 10 min, 4°C and $12,000 \times g$, 35 min, 4°C). The supernatant (cytoplasmic fraction) and pellet (mitochondrial fraction) were collected, and the pellet was further washed with wash buffer (800 mM KCl, 10 mM EDTA, 5 mM $MgCl_2$, and 20 mM HEPES (pH 7.4), and the protease inhibitor) three times and yielded the final mitochondrial fraction. To confirm that pure extracts were obtained, the mitochondrial, nuclear, and cytoplasmic fractions were separated by SDS–PAGE, and the presence of mitochondrial VDAC1, BCL2, nuclear Histone H3, and cytoplasmic GAPDH was detected by immunoblot.

### Isolation of submitochondrial fractions

Six mitochondrial fraction samples were divided into three groups, with two samples in each group. The first group was resuspended in 300 µl of homogenization buffer, the second group in 300 µl of hypotonic swelling buffer (10 mM HEPES/KOH, pH 7.4, 1 mM EDTA), and the third group in 300 µl of homogenization buffer supplemented with 0.5% (vol/vol) Triton X-100, followed by a 10-min incubation on ice. Subsequently, one sample from each group was exposed to proteinase K (70 µg/ml) for 20 min on ice, while the other sample was kept untreated as a control. Following the treatments, mitochondrial proteins were precipitated using 300 µl of 30% TCA (wt/vol) and incubated on ice for 10 min. The proteins were collected by centrifugation at $18,000 \times g$ for 10 min at 4°C, washed with 1 ml of 100% ethanol, and centrifuged again. The resulting pellets were dissolved in 100 µl of SDS Lysis Buffer, boiled at 105°C for 8 min, and subjected to WB analyses.

### CRISPR/Cas9-mediated gene KO cell lines

C4-2B or Huh-7 cells were plated and transfected with PX459 plasmids overnight. 24 hr after transfection, 1 µg/ml puromycin was used to screen cells for 3 days. Living cells were seeded in a 96-well plate by limited dilution to isolate a monoclonal cell line. The knockout cell clones are screened by WB and validated by Sanger sequencing. Sequences of gene-specific sgRNAs are listed in *Supplementary file 4*.

### RT-qPCR assays

Total RNA from cells was extracted by using TRIzol reagent (TIANGEN), followed by reverse transcription into cDNA using the HiScript III First Strand cDNA Synthesis Kit (Vazyme). The synthesized cDNAs

were then subjected to PCR amplification using ChamQ SYBR qPCR Master Mix (Vazyme) in CFX Real-Time PCR system (Bio-Rad). The relative mRNA levels of *NOXA* were quantified using the $2^{-\Delta\Delta CT}$ method with normalization to *GAPDH*. The primer sequences are listed in **Supplementary file 4**.

### In vivo ubiquitination assay I

293T cells were transfected with HA-ubiquitin and indicated constructs. After 36 hr, cells were treated with MG132 (30 µM) for 6 hr and then lysed in 1% SDS buffer (Tris [pH 7.5], 0.5 mM EDTA, 1 mM DTT) and boiled for 10 min. For immunoprecipitation, the cell lysates were diluted 10-fold in Tris-HCl buffer and incubated with anti-NOXA or IgG-conjugated beads (Sigma) for 4 hr at 4°C. The bound beads are then washed four times with BC100 buffer (20 mM Tris-HCl, pH 7.9,100 mM NaCl, 0.2 mM EDTA, 20% glycerol) containing 0.2% Triton X-100. The proteins were eluted with FLAG peptide for 2 hr at 4°C. The ubiquitinated form of NOXA was detected by WB using anti-HA antibody.

### In vivo ubiquitination assay II

293T cells were co-transfected with (His)$_6$-tagged ubiquitin and the indicated constructs. After 24 hr, cells were lysed in buffer A (6 M guanidine-HCl, 0.1 M Na$_2$HPO$_4$/NaH$_2$PO$_4$, 10 mM imidazole (pH 8.0)). After sonication, the cell lysates were incubated with Ni–NTA beads (QIAGEN) for 3 hr at room temperature (RT). Subsequently, the pull-down products were washed once with buffer A, twice with buffer A/TI (buffer A:buffer TI = 1:3), and once with buffer TI (25 mM Tris-HCl and 20 mM imidazole; (pH 6.8)). Pull-down proteins and WCL were detected by WB.

## IF and confocal microscopy

HeLa cells were seeded on glass coverslips in 12-well plates and harvested at 70% confluence. The cells were washed with PBS and fixed with 4% paraformaldehyde in PBS. After permeabilization with 0.3% Triton X-100 for 5 min and then in the blocking solution (PBS plus 5% donkey serum), for 1 hr at RT. The cells were then incubated with primary antibodies at 4°C overnight. After washing with PBST buffer, fluorescence-labeled secondary antibodies were applied. DAPI was utilized to stain nuclei. The glass coverslips were mounted on slides and imaged using a confocal microscope (LSM880, Zeiss) with a 63×/1.4 NA Oil PSF Objective. Quantitative analyses were performed using ImageJ software.

For mouse tissues staining, the mouse tissues were isolated from mice after perfusion with 0.1 M PBS (pH 7.4) and fixed for 3 days with 4% PFA at 4°C. The tumor tissues were then placed in a 30% sucrose solution for 5 days for dehydration. The tumors were embedded into the OCT block and frozen for cryostat sectioning. Cryostat sections (45 µm thick) were washed with PBS, and then incubated in blocking solution (PBS containing 10% goat serum, 0.3% Triton X-100, pH 7.4) for 2 h at RT. The samples were stained with primary antibodies overnight at 4°C, after washing with PBST buffer, fluorescence-labeled secondary antibodies were applied at RT for 2 hr. DAPI was utilized to stain nuclei. The sections were then sealed with an anti-fluorescence quencher. The samples were visualized and imaged using a confocal microscope (LSM880, Zeiss) with a 63×/1.4 NA Oil PSF Objective. Quantitative analyses were performed using ImageJ software.

## Apoptosis assays

Annexin V-FITC (fluorescein isothiocyanate) and propidium iodide (PI) double staining (Dojindo) were used to detect the apoptosis rates. The cells were cultured in 6-well plates at a density of $1.2 \times 10^5$/well and allowed to adhere to the culture plate overnight. Then the medium was replaced with fresh medium containing indicated drugs for a certain time. The cells were then trypsinized by EDTA-free trypsin and washed twice with cold PBS. Aliquots of the cells were resuspended in 100 µl of binding buffer and stained with 5 µl of annexin V-FITC and 5 µl of PI working solution for 15 min at RT in the dark. All flow cytometry analyses were carried out using a Fortessa flow cytometer (BD Bioscience). The subsequent data analysis was conducted using FlowJo software.

## Generation and breeding of *Wsb2* KO mice

Mice with murine Wsb2 KO were designed and generated from Shanghai Model Organisms Center (Shanghai, China). In brief, the CRISPR/Cas9 system was microinjected into the fertilized eggs of C57BL/6JGpt mice. Fertilized eggs were transplanted to obtain positive F0 mice, which were confirmed by PCR and sequencing. A stable F1 generation mouse model was obtained by mating

positive F0 generation mice with C57BL/6JGpt mice. The genotype of F1 mice was identified by PCR and confirmed by sequencing. The sequences used for CRISPR-Cas9 editing and the primers used for genotyping are listed in *Supplementary file 4*.

Mice were maintained under a 12 hr/12 hr light/dark cycle at 22–25°C and 40–50% humidity with standard food and water available ad libitum. All procedures for animal care and animal experiments were carried out in accordance with the guidelines of the Care and Use of Laboratory Animals proposed by the Institute of Development Biology and Molecular Medicine and Shanghai Municipality, PR China. The male C57BL/6JGpt mice (8 months old) were divided into 2 groups ($Wsb2^{+/+}$ and $Wsb2^{-/-}$; $n$ = 5/group), and each group was given by gavage of ABT-199 (100 mg/kg/day) or vehicle (10% β-cyclodextrin) for 1 week. Then, we collect blood samples from the tail vein of mice before and after oral administration to measure indicators of myocardial zymogram using an Olympus AU640 automatic biochemical analyzer (Olympus).

## MEFs generation and immortalization

Timed pregnant female mice at embryonic days 12.5–14.5 were sacrificed, and the embryos were carefully dissected to remove the cerebrum, internal organs, and limbs. The remaining tissues were cut into small pieces and treated with trypsin-EDTA (0.25%) for 10 min at 37°C. The trypsin was neutralized with DMEM, a complete medium supplemented with 10% FBS and 1% penicillin/streptomycin. The culture media were changed every 2–3 days until the cells reached confluence. To immortalize MEFs, they were passaged up to approximately 10 times before infection with lentiviral vectors expressing the SV40 large T-antigen. Stable transduction was achieved with puromycin selection. The successful integration of the immortalizing gene was confirmed through Sanger sequencing and WB analysis.

## Mouse tumor implantation

All experimental protocols were approved in advance by the Ethics Review Committee for Animal Experimentation of Fudan University. Four- to six-week-old male BALB/c nu/nu mice obtained from SLAC Laboratory Animal Co, Ltd were bred and maintained in our institutional pathogen-free mouse facilities. Mice were randomly divided into four groups ($n$ = 6/group): vehicle (distilled water); ABT-737 (20 mg/kg); R8-C-ter (20 mg/kg); and ABT-737+R8-C-ter. Huh-7 tumors were established by subcutaneously injecting $5 \times 10^6$ Huh-7 cells in 100 μl of PBS buffer into the right flank of 6-week-old nude mice. After 1 week, vehicle and indicated drug treatments were administered once daily by intraperitoneal injection (i.p). At the end of 3 weeks, mice were killed and in vivo solid tumors were dissected and weighed.

## Pan-cancer dataset acquisition and analysis

Pan-cancer gene expression analysis based on tumor and normal samples was derived from the TCGA (transcriptome datasets, http://gepia2.cancer-pku.cn/). Public databases (TCGA) were used to analyze the correlations of WSB2 expression with clinical risk factors. Additional publicly available RNA-seq datasets (GSE104580 and GSE109211) provided for sorafenib-response and sorafenib-non-response HCC patients.

## IHC analysis

A total of 84 patients with localized PRAD, who underwent radical prostatectomy between January 2007 and July 2014 at Fudan University Shanghai Cancer Center (FUSCC), were included in this study. All the patients underwent regular postoperative reviews and had long-term follow-up data. This study was in accordance with the recommendations of the Research Ethics Committee of FUSCC according to the provisions of the Declaration of Helsinki (as revised in Fortaleza, Brazil, October 2013). The protocol was approved by the Research Ethics Committee of FUSCC. Informed consent for the use of clinical data was obtained from all the patients recruited in this study. The TMA consists of 29 LIHC patient specimens obtained from Shanghai Biochip Co, Ltd (Shanghai). To confirm the specificity of the anti-WSB2 antibody, we conducted genetic control for the IHC analysis using an anti-WSB2 antibody in both parental and WSB2 KO C4-2B cells.

## Statistical analysis

Statistical analysis was performed using GraphPad Prism (GraphPad Software), and the differences between the two groups were analyzed using one-way analysis of variance (ANOVA) or two-way

ANOVA. All data were displayed as means ± SE values for experiments conducted with at least three replicates. * represents $p < 0.05$; ** represents $p < 0.01$; *** represents $p < 0.001$, **** represents $p < 0.0001$.

## Acknowledgements

This work was in part supported by the National Natural Science Foundation of China (Nos. 92357301, 32370726, and 91957125 to CW; 82272992, 91954106, and 81872109 to KG; 82270415 to LW; 81902614 to KC), the State Key Development Programs of China (No. 2022YFA1104200 to CW), the Natural Science Foundation of Shanghai (No. 22ZR1406600 to CW; 22ZR1449200 to KG, 22ZR1448600 to YX), Science and Technology Research Program of Shanghai (No. 9DZ2282100); Open Research Fund of the State Key Laboratory of Genetics and Development of Complex Phenotypes, Fudan University (No. SKLGE-2111 to KG), Science and Technology Research Program of Shanghai (No. 9DZ2282100), Central Guidance on Local Science and Technology Development Foundation (No. 2021ZY0037 to RM), and China Postdoctoral Science Foundation (No. GZC20240296 to YJC).

## Additional information

### Funding

| Funder | Grant reference number | Author |
| --- | --- | --- |
| National Natural Science Foundation of China | 92357301 | Chenji Wang |
| National Natural Science Foundation of China | 32370726 | Chenji Wang |
| National Natural Science Foundation of China | 91957125 | Chenji Wang |
| National Natural Science Foundation of China | 82272992 | Kun Gao |
| National Natural Science Foundation of China | 82270415 | Lixin Wang |
| National Natural Science Foundation of China | 91954106 | Kun Gao |
| National Natural Science Foundation of China | 81872109 | Kun Gao |
| National Natural Science Foundation of China | 81902614 | Kun Chang |
| State Key Development Programs of China | 2022YFA1104200 | Chenji Wang |
| Natural Science Foundation of Shanghai Municipality | 22ZR1406600 | Chenji Wang |
| Natural Science Foundation of Shanghai Municipality | 22ZR1449200 | Kun Gao |
| Natural Science Foundation of Shanghai Municipality | 22ZR1448600 | Yaoting Xu |
| Science and Technology Research Program of Shanghai | 9DZ2282100 | Chenji Wang |

| Funder | Grant reference number | Author |
| --- | --- | --- |
| Open Research Fund of the State Key Laboratory of Genetics and Development of Complex Phenotypes | SKLGE-2111 | Kun Gao |
| Central Guidance on Local Science and Technology Development Foundation | 2021ZY0037 | Mo Ren |
| China Postdoctoral Science Foundation | GZC20240296 | Yingji Chen |

The funders had no role in study design, data collection, and interpretation, or the decision to submit the work for publication.

## Author contributions

Dongyue Jiao, Data curation, Validation, Investigation, Visualization, Writing - original draft; Kun Chang, Validation, Investigation, Methodology; Jiamin Jin, Yingji Chen, Mo Ren, Yucong Zhang, Investigation; Kun Gao, Supervision, Funding acquisition, Investigation, Writing - original draft; Yaoting Xu, Funding acquisition, Investigation, Project administration, Resources; Lixin Wang, Supervision, Methodology, Project administration; Chenji Wang, Conceptualization, Resources, Data curation, Supervision, Funding acquisition, Investigation, Visualization, Methodology, Writing - original draft, Project administration, Writing - review and editing

## Author ORCIDs

Dongyue Jiao (ID) https://orcid.org/0009-0002-4061-0578
Chenji Wang (ID) https://orcid.org/0000-0002-5752-6439

## Ethics

This study was in accordance with the recommendations of the Research Ethics Committee of FUSCC according to the provisions of the Declaration of Helsinki (as revised in Fortaleza, Brazil, October 2013). The protocol was approved by the Research Ethics Committee of FUSCC. Informed consent for the use of clinical data was obtained from all the patients recruited in this study.
All procedures for animal care and animal experiments were carried out in accordance with the guidelines of the Care and Use of Laboratory Animals proposed by Fudan University (Permit Number: IDM2024050).

Reviewer #1 (Public Review): https://doi.org/10.7554/eLife.98372.3.sa1
Author response https://doi.org/10.7554/eLife.98372.3.sa2

# Additional files

## Supplementary files

Supplementary file 1. Top 100 co-dependent genes of WSB2.

Supplementary file 2. The GO analysis of the top 500 co-dependent genes of WSB2.

Supplementary file 3. The DepLink analysis of top correlated drugs with WSB2.

Supplementary file 4. Sequence information.

MDAR checklist

## Data availability

All data generated or analysed during this study are included in the manuscript and supporting files.

The following previously published datasets were used:

| Author(s) | Year | Dataset title | Dataset URL | Database and Identifier |
|---|---|---|---|---|
| Pinyol R, Montal RR, Moeini AA, Llovet JM | 2020 | Using a novel gene signature for predicting the efficacy of the transarterial chemoembolization in patients with hepatocellular carcinoma | https://www.ncbi.nlm.nih.gov/geo/query/acc.cgi?acc=GSE104580 | NCBI Gene Expression Omnibus, GSE104580 |
| Hui KM, Shi M | 2018 | Molecular predictors of prevention of recurrence in hepatocellular carcinoma with sorafenib as adjuvant treatment in the phase 3 STORM trial | https://www.ncbi.nlm.nih.gov/geo/query/acc.cgi?acc=GSE109211 | NCBI Gene Expression Omnibus, GSE109211 |
| Wainberg M, Kamber RA, Balsubramani A, Meyers RM, Sinnott-Armstrong N, Hornburg D, Jiang L, Chan J, Jian R, Gu M, Shcherbina A, Dubreuil MM, Spees K, Meuleman W, Snyder MP, Bassik MC, Kundaje A | 2019 | Molecular predictors of prevention of recurrence in hepatocellular carcinoma with sorafenib as adjuvant treatment in the phase 3 STORM trial | https://mitra.stanford.edu/bassik/michael/cluster_heatmaps | mitra.stanford.edu, module#528 |

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

# Appendix 1

## Appendix 1—key resources table

| Reagent type (species) or resource | Designation | Source or reference | Identifiers | Additional information |
|---|---|---|---|---|
| Antibody | anti-WSB2 (Rabbit polyclonal) | Abclonal | Cat# WG-05341D, RRID:AB_3674706 | IHC(1:200) WB(1:1000) |
| Antibody | anti-CUL5 (Rabbit polyclonal) | Abclonal | Cat A5369 RRID:AB_2766179 | WB(1:1000) |
| Antibody | anti-RBX2 (Rabbit polyclonal) | Proteintech | Cat# 11905–1-AP RRID:AB_10697836 | WB(1:1000) |
| Antibody | anti-ELOB (Rabbit polyclonal) | Abclonal | Cat A5362 RRID:AB_2766172 | WB(1:1000) |
| Antibody | anti-ELOC (Rabbit polyclonal) | Abclonal | Cat# A12515 RRID:AB_2759355 | WB(1:1000) |
| Antibody | anti-NOXA (Rabbit monoclonal) | CST | Cat# 14766 S RRID:AB_2798602 | WB(1:1000) |
| Antibody | anti-Noxa (Rabbit monoclonal) | SANTA CRUZ | Cat# sc-56169 RRID:AB_784877 | WB(1:1000) |
| Antibody | anti-BAX (Rabbit monoclonal) | Abclonal | Cat# A19684 RRID:AB_2862733 | WB(1:1000) |
| Antibody | anti-BAK1 (Rabbit monoclonal) | Abclonal | Cat# A10754 RRID:AB_2758197 | WB(1:1000) |
| Antibody | anti-MCL-1 (Rabbit polyclonal) | Proteintech | Cat# 16225–1-AP RRID:AB_2143977 | WB(1:1000) |
| Antibody | anti-BCL-2 (Rabbit monoclonal) | CST | Cat 4223T RRID:AB_1903909 | WB(1:1000) |
| Antibody | anti-BCL-W (Rabbit polyclonal) | Abclonal | Cat# A13471 RRID:AB_2760333 | WB(1:1000) |
| Antibody | anti-BCL-XL (Rabbit monoclonal) | Abclonal | Cat# A19703 RRID:AB_2862745 | WB(1:1000) |
| Antibody | anti-BAD (Rabbit monoclonal) | Abclonal | Cat# A19595 RRID:AB_2862688 | WB(1:1000) |
| Antibody | anti-MARCH5 (Rabbit polyclonal) | Proteintech | Cat# 12213–1-AP RRID:AB_10638602 | WB(1:1000) |
| Antibody | anti-TOM70 (Rabbit monoclonal) | CST | Cat# 65,619T RRID:AB_3411916 | WB(1:1000) |
| Antibody | anti-SMAC (Rabbit monoclonal) | CST | Cat# 15,108T RRID:AB_2798711 | WB(1:1000) |
| Antibody | anti-HSP60 (Rabbit monoclonal) | CST | Cat# 12,165T RRID:AB_2636980 | WB(1:1000) |
| Antibody | anti-Histone H3 (Rabbit polyclonal) | CST | Cat# 9715 S RRID:AB_331563 | WB(1:1000) |
| Antibody | anti-cleaved PARP1 (Rabbit monoclonal) | CST | Cat# 5625T RRID:AB_10699459 | IF(1:200) |
| Antibody | anti-CASP9 (Rabbit polyclonal) | CST | Cat# 9502 S RRID:AB_2068621 | WB(1:1000) |
| Antibody | anti-CASP7 (Rabbit polyclonal) | CST | Cat# 9492 S RRID:AB_2228313 | WB(1:1000) |
| Antibody | anti-cleaved CASP7 (Rabbit monoclonal) | CST | Cat# 8438 S RRID:AB_11178377 | IF(1:200) |
| Antibody | anti-CASP3 (Rabbit monoclonal) | CST | Cat# 9668 S RRID:AB_2069870 | WB(1:1000) |
| Antibody | anti-cleaved CASP3 (Rabbit monoclonal) | CST | Cat# 9664 S RRID:AB_2070042 | IF(1:200) WB(1:1000) |
| Antibody | anti-GFP (Mouse monoclonal) | Abclonal | Cat# AE012 RRID:AB_2770402 | WB(1:1000) |
| Antibody | anti-FLAG (Mouse polyclonal) | MBL | Cat# PM020 RRID:AB_591224 | WB(1:1000) |
| Antibody | anti-Myc (Mouse monoclonal) | MBL | Cat# M192-3 RRID:AB_11160947 | WB(1:1000) |
| Antibody | anti-HA (Mouse monoclonal) | MBL | Cat# M180-3 RRID:AB_10951811 | WB(1:1000) |
| Antibody | anti-GAPDH (Mouse monoclonal) | Abcam | Cat# ab8245 RRID:AB_2107448 | WB(1:1000) |
| Antibody | anti-β-Actin (Rabbit monoclonal) | Abclonal | Cat# AC026 RRID:AB_2768234 | WB(1:1000) |
| Peptide, recombinant protein | FLAG peptide | ChinaPeptides | Cat# 04010006736 | |
| Peptide, recombinant protein | Penicillin-Streptomycin | Invitrogen | Cat# 15070063 | |
| Commercial assay or kit | KOD-Plus-Mutagenesis Kit | TOYOBO | Cat# SMK-101 | |
| Commercial assay or kit | Annexin V-FITC Apoptosis Detection Kit | Dojindo | Cat# AD10 | |
| Commercial assay or kit | TUNEL staining Kit | Beyotime | Cat# C1086 | |

*Appendix 1 Continued on next page*

*Appendix 1 Continued*

| Reagent type (species) or resource | Designation | Source or reference | Identifiers | Additional information |
|---|---|---|---|---|
| Chemical compound, drug | L-Glutamine | Gibco | Cat# 25030149 | |
| Chemical compound, drug | MG132 | Selleckchem | Cat# S2619 | |
| Chemical compound, drug | ABT-737 | MCE | Cat# HY-50907 | |
| Chemical compound, drug | ABT-199 | MCE | Cat# HY-15531 | |
| Chemical compound, drug | BAY-1143572 | Selleck | Cat# S8727 | |
| Chemical compound, drug | AZD5991 | Selleck | Cat# S8643 | |
| Chemical compound, drug | Cycloheximide | Sigma | Cat# 66-81-9 | |
| Chemical compound, drug | Puromycin | Sigma | Cat# P8833 | |
| Chemical compound, drug | Protease Inhibitor Cocktail (EDTA-Free, 100 X in DMSO) | Selleck | Cat# B14001 | |
| Chemical compound, drug | Trizol | Thermo Fisher | Cat# 14496026 | |
| Chemical compound, drug | EZ Trans | Shanghai Life-iLab Biotech | Cat# AC04L092 | |
| Chemical compound, drug | EndoFectinTM-MAX | iGeneBio | Cat# EF013 | |
| Chemical compound, drug | ChamQ SYBR qPCR Master Mix | Vazyme Biotech | Cat# Q311 | |
| Chemical compound, drug | Phanta Max Super-Fidelity DNA Polymerase | Vazyme Biotech | Cat# P505 | |
| Chemical compound, drug | Anti-FLAG M2 | Sigma | Cat# SLCD1942 | |

