## [Editor Report · eLife Assessment]

This study reports a **fundamental** observation concerning cell death regulation by the anti-apoptotic BCL2 family NOXA. The authors **convincingly** demonstrate that NOXA is destabilized through the interaction with WSB2, a substrate receptor in CRL5 ubiquitin ligase complex, sensitizing the cells to treatments. These are key findings for cell biologists and cancer researchers as they identified a new target impacting drug responsiveness in cancer therapies.

---

## [Referee Report · Reviewer #1 (Public Review)]

Summary:

In this manuscript, Jiao D et al reported the induction of synthetic lethality by combined inhibition of anti-apoptotic BCL-2 family proteins and WSB2, a substrate receptor in CRL5 ubiquitin ligase complex. Mechanistically, WSB2 interacts with NOXA to promote its ubiquitylation and degradation. Cancer cells deficient in WSB2, as well as heart and liver tissues from Wsb2-/- mice exhibit high susceptibility to apoptosis induced by inhibitors of BCL-2 family proteins. The anti-apoptotic activity of WSB2 is partially dependent on NOXA.

Overall, the finding that WSB2 disruption triggers synthetic lethality to BCL-2 family protein inhibitors by destabilizing NOXA is rather novel. The manuscript is largely hypothesis-driven, with experiments that are adequately designed and executed. However, there are quite a few issues for the authors to address, including those listed below.

Specific comments from the previous round of review:

(1) At the beginning of the Results section, a clear statement is needed as to why the authors are interested in WSB2 and what brought them to analyze "the genetic co-dependency between WSB2 and other proteins".

(2) In general, the biochemical evidence supporting the role of WSB2 as a SOCS box-containing substrate-binding receptor of CRL5 E3 in promoting NOXA ubiquitylation and degradation is relatively weak. First, since NOXA2 binds to WSB2 on its SOCS box, which consists of a BC box for Elongin B/C binding and a CUL5 box for CUL5 binding, it is crucial to determine whether the binding of NOXA on the SOCS box affects the formation of CRL5WSB2 complex. The authors should demonstrate the endogenous binding between NOXA and the CRL5WSB2 complex. Additionally, the authors may also consider manipulating CUL5, SAG, or ElonginB/C to assess if it would affect NOXA protein turnover in two independent cell lines. Second, in all the experiments designed to detect NOXA ubiquitylation in cells, the authors utilized immunoprecipitation (IP) with FLAG-NOXA/NOXA, followed by immunoblotting (IB) with HA-Ub. However, it is possible that the observed poly-Ub bands could be partly attributed to the ubiquitylation of other NOXA binding proteins. Therefore, the authors need to consider performing IP with HA-Ub and subsequently IB with NOXA. Alternatively, they could use Ni-beads to pull down all His-Ub-tagged proteins under denaturing conditions, followed by the detection of FLAG-tagged NOXA using anti-FLAG Ab. The authors are encouraged to perform one of these suggested experiments to exclude the possibility of this concern. Furthermore, an in vitro ubiquitylation assay is crucial to conclusively demonstrate that the polyubiquitylation of NOXA is indeed mediated by the CRL5WSB2 complex.

(3) In their attempt to map the binding regions between NOXA and WSB2, the authors utilized exogenous proteins of both WSB2 and NOXA. To strengthen their findings, it would be more convincing to perform IP with exogenous wt/mutant WSB2 or NOXA and subsequently perform IB to detect endogenous NOXA or WSB2, respectively. Additionally, an in vitro binding assay using purified proteins would provide further evidence of a direct binding between NOXA and WSB2.

Comments on latest version:

The authors have adequately addressed my previous comments.

---

## [Author Response]

The following is the authors’ response to the original reviews.

**Public Reviews:**

**Reviewer #1 (Public Review):**
Summary:I In this manuscript, Jiao D et al reported the induction of synthetic lethal by combined inhibition of anti-apoptotic BCL-2 family proteins and WSB2, a substrate receptor in CRL5 ubiquitin ligase complex. Mechanistically, WSB2 interacts with NOXA to promote its ubiquitylation and degradation. Cancer cells deficient in WSB2, as well as heart and liver tissues from Wsb2-/- mice exhibit high susceptibility to apoptosis induced by inhibitors of BCL-2 family proteins. The anti-apoptotic activity of WSB2 is partially dependent on NOXA.Overall, the finding, that WSB2 disruption triggers synthetic lethality to BCL-2 family protein inhibitors by destabilizing NOXA, is rather novel. The manuscript is largely hypothesis-driven, with experiments that are adequately designed and executed. However, there are quite a few issues for the authors to address, including those listed below.Specific comments:(1) At the beginning of the Results section, a clear statement is needed as to why the authors are interested in WSB2 and what brought them to analyze "the genetic co-dependency between WSB2 and other proteins".

We thank the reviewer for raising this important point. We agree that a clear rationale should be provided at the beginning of the Results section. As reported in previous studies [Ref: 1, 2, 3], strong synthetic interactions have been observed between WSB2 and several mitochondrial apoptosis-related factors, including MCL-1, BCL-xL, and MARCH5. We have referenced these findings in the Discussion section. Motivated by these studies, we became interested in the role of WSB2 and aimed to investigate the specific mechanisms underlying its synthetic lethality with anti-apoptotic BCL-2 family members. We will revise the beginning of the Results section to clearly state this rationale.

(1) McDonald, E.R., 3rd et al. Project DRIVE: A Compendium of Cancer Dependencies and Synthetic Lethal Relationships Uncovered by Large-Scale, Deep RNAi Screening. Cell 170, 577-592 e510 (2017).

(2) DeWeirdt, P.C. et al. Genetic screens in isogenic mammalian cell lines without single cell cloning. Nat Commun 11, 752 (2020).

(3) DeWeirdt, P.C. et al. Optimization of AsCas12a for combinatorial genetic screens in human cells. Nat Biotechnol 39, 94-104 (2021).

(2) In general, the biochemical evidence supporting the role of WSB2 as a SOCS box-containing substrate-binding receptor of CRL5 E3 in promoting NOXA ubiquitylation and degradation is relatively weak. First, since NOXA binds to WSB2 on its SOCS box, which consists of a BC box for Elongin B/C binding and a CUL5 box for CUL5 binding, it is crucial to determine whether the binding of NOXA on the SOCS box affects the formation of CRL5WSB2 complex. The authors should demonstrate the endogenous binding between NOXA and the CRL5WSB2 complex. Additionally, the authors may also consider manipulating CUL5, SAG, or ElonginB/C to assess if it would affect NOXA protein turnover in two independent cell lines.

We thank the reviewer for raising this important point. To determine whether endogenous NOXA binds to the intact CRL5^WSB2^ complex, we performed co-immunoprecipitation assays using an antibody against NOXA. Indeed, NOXA co-immunoprecipitated with all subunits of the CRL5^WSB2^ complex (Figure 2—figure supplement 1D), suggesting that NOXA binding to WSB2 does not disrupt interactions between WSB2 and the other CRL5 subunits. Moreover, depletion of CRL5 complex components (RBX2/SAG, CUL5, ELOB, or ELOC) through siRNAs in C4-2B or Huh-7 cells also resulted in a marked increase in NOXA protein levels.

Second, in all the experiments designed to detect NOXA ubiquitylation in cells, the authors utilized immunoprecipitation (IP) with FLAG-NOXA/NOXA, followed by immunoblotting (IB) with HA-Ub. However, it is possible that the observed poly-Ub bands could be partly attributed to the ubiquitylation of other NOXA binding proteins. Therefore, the authors need to consider performing IP with HA-Ub and subsequently IB with NOXA. Alternatively, they could use Ni-beads to pull down all His-Ub-tagged proteins under denaturing conditions, followed by the detection of FLAG-tagged NOXA using anti-FLAG Ab. The authors are encouraged to perform one of these suggested experiments to exclude the possibility of this concern. Furthermore, an in vitro ubiquitylation assay is crucial to conclusively demonstrate that the polyubiquitylation of NOXA is indeed mediated by the CRL5WSB2 complex.

We appreciate the reviewer for raising these important considerations regarding our ubiquitylation assays. We fully acknowledge the reviewer's concern that classical ubiquitination assays could potentially detect ubiquitination of proteins interacting with NOXA. However, we would like to clarify that our experimental conditions effectively mitigate this issue. Specifically, cells were lysed using buffer containing 1% SDS followed by boiling at 105°C for 5 minutes. These rigorous denaturing conditions ensure disruption of non-covalent protein interactions, thereby effectively eliminating the possibility of detecting ubiquitination signals from NOXA-associated proteins.

Regarding the suggestion to perform an in vitro ubiquitination assay, we agree this experiment would indeed provide additional evidence. However, due to significant technical complexities associated with reconstituting CRL5-based E3 ubiquitin ligase activity in vitro—which would require the expression and purification of at least six recombinant proteins—such experiments are rarely performed in this context. Furthermore, NOXA is uniquely localized as a membrane protein on the mitochondrial outer membrane, posing additional significant challenges for protein expression and purification. Given the robustness of our current in vivo ubiquitylation assay under stringent denaturing conditions, we believe our existing data sufficiently and conclusively demonstrate NOXA ubiquitination mediated by the CRL5^WSB2^ complex.

(3) In their attempt to map the binding regions between NOXA and WSB2, the authors utilized exogenous proteins of both WSB2 and NOXA. To strengthen their findings, it would be more convincing to perform IP with exogenous wt/mutant WSB2 or NOXA and subsequently perform IB to detect endogenous NOXA or WSB2, respectively. Additionally, an in vitro binding assay using purified proteins would provide further evidence of a direct binding between NOXA and WSB2.

We thank the reviewer for raising these important issues. In response to the reviewer’s suggestion to map the binding regions between NOXA and WSB2 more convincingly, we have indeed performed semi-endogenous Co-IP assays, which yielded results consistent with our exogenous protein experiments (Figure 3—figure supplement 1A, B). Concerning the recommendation to further validate direct interaction using purified recombinant proteins, we encountered substantial technical difficulties in obtaining pure and soluble recombinant WSB2 protein. Additionally, given that NOXA is an outer mitochondrial membrane protein and the interaction occurs on mitochondria, we believe that an in vitro binding assay may have limited physiological relevance. We hope the reviewer can appreciate these practical challenges and our current evidence supporting the strong interaction between NOXA and WSB2.

**Reviewer #2 (Public Review):**
Summary:Exploring the DEP-MAP database and two drug-screen databases, the authors identify WSB2 as an interactor of several BCL2 proteins. In follow-up experiments, they show that CRL5/WSB2 controls NOXA protein levels via K48 ubiquitination following direct protein-protein interaction, and cell death sensitivity in the context of BH3 mimetic treatment, where WSB2 depletion synergizes with drug treatment.Strengths:The authors use a set of orthogonal methods across different model cell lines and a new WSB2 KO mouse model to confirm their findings. They also manage to correlate WSB2 expression with poor prognosis in prostate and liver cancer, supporting the idea that targeting WSB2 may sensitize cancers for treatment with BH3 mimetics.Weaknesses:The conclusions drawn based on the findings in cancer patients are very speculative, as regulation of NOXA cannot be the sole function of CRL5/WSB2 and it is hence unclear what causes correlation with patient survival. Moreover, the authors do not provide a clear mechanistic explanation of how exactly higher levels of NOXA promote apoptosis in the absence of WSB2. This would be important knowledge, as usually high NOXA levels correlate with high MCL1, as they are turned over together, but in situations like this, or loss of other E3 ligases, such as MARCH, the buffering capacity of MCL1 is outrun, allowing excess NOXA to kill (likely by neutralizing other BCL2 proteins it usually does not bind to, such as BCLX). Moreover, a necroptosis-inducing role of NOXA has been postulated. Neither of these options is interrogated here.
**Recommendations For The Authors:**

**Reviewer #1 (Recommendations For The Authors):**
(1) Figure 2J. The authors showed that "the mRNA levels of NOXA were even reduced in WSB2-KO cells compared to parental cells". What is the possible mechanism? This point should at least be discussed.

We thank the reviewer for raising these important issues. The underlying mechanisms for the significantly lower mRNA levels of NOXA following the KO of *WSB2* are not fully understood at present. However, we propose that this could represent a form of negative feedback regulation at the level of gene expression. Specifically, when the protein levels of BNIP3/3L rise sharply, it may activate mechanisms that suppress their own mRNA synthesis or stability, serving as a buffering system to prevent further protein accumulation. Such negative feedback loops may be critical for maintaining cellular homeostasis and avoiding excessive protein production. Moreover, this phenomenon is frequently observed in other studies investigating substrates targeted by E3 ubiquitin ligases for degradation. We have elaborated on this point in the Discussion section.

(2) Figure 2M. A previous study has clearly demonstrated that NOXA is subjected to ubiquitylation and degradation by CRL5 E3 ligase (PMID: 27591266). This paper should be cited. Also, in that publication, NOXA ubiquitylation is via the K11 linkage, not the K48 linkage. The authors should include K11R mutant in their assay.

We thank the reviewer for raising this important issue. We thank the reviewer for suggesting the relevant reference (PMID: 27591266), which we have now cited accordingly. Additionally, we would like to clarify that our new in vivo ubiquitination assays included the K11R and K11-only ubiquitin mutants, and our data demonstrate that WSB2-mediated NOXA ubiquitination indeed involves the K11 linkage ubiquitination (Figure 2—figure supplement 1E).

(3) Figure 3H, J. The authors stated, "By mutating these lysine residues to arginine, we found that WSB2-mediated NOXA ubiquitination was completely abolished". Which one of the three lysine residues is playing the dominant role?

We thank the reviewer for raising this important issue. To address this, we generated FLAG-NOXA mutants individually substituting lysine residues K35, K41, and K48 with arginine. In vivo ubiquitination assays demonstrated that lysine 48 (K48) is the predominant residue responsible for WSB2-mediated NOXA ubiquitination (Figure 3—figure supplement 1C).

(4) Figure 3N. The authors need to show that the fusion peptide containing C-terminal NOXA peptide competitively inhibits the interaction between endogenous WSB2 and NOXA and extends the protein half-life of NOXA, leading to NOXA accumulation.

We sincerely thank the reviewer for raising these important issues. As suggested, we investigated whether the fusion peptide containing the C-terminal NOXA sequence competitively disrupts the interaction between endogenous WSB2 and NOXA, subsequently influencing NOXA stability. Our results demonstrated that treatment with this fusion peptide indeed significantly reduced the endogenous interaction between WSB2 and NOXA (Figure 3—figure supplement 1D). Furthermore, we observed that the peptide dose-dependently increased endogenous NOXA protein levels and prolonged its protein half-life, thereby resulting in the accumulation of NOXA (Figure 3N; Figure 3—figure supplement 1E, F). These findings collectively indicate that the fusion peptide competitively inhibits the WSB2-NOXA interaction, stabilizes NOXA protein, and enhances its accumulation.

(5) Figure 4. (a) It would be better to investigate whether WSB2 knockdown can sensitize cancer cells to the treatment with ABT-737 or AZD5991, evidenced by a decrease in both IC50 values and clonogenic survival rates and whether such sensitization is dependent on NOXA. (b) The authors need to show the levels of cleaved caspase-3/7/9 and the percentages of apoptotic cells in shNC cells upon silencing of WSB2 in Figure 4A-F. (c) It will be more convincing to repeat the experiment to show synthetic lethality by WSB2 disruption and MCL-1 inhibitor AZD5991 treatment using another cell line, such as WSB2-deficient Huh-7 cells in Figure 4 I&J.

We sincerely thank the reviewer for these valuable and constructive suggestions.Regarding point (a): We believe that our current Western blot and flow cytometry data (Figure 4G–L) have already provided strong evidence that WSB2 depletion enhances apoptosis in response to ABT-737 and AZD5991. Therefore, we consider that additional IC50 and clonogenic survival assays, while informative, may not be essential for supporting our conclusion. Furthermore, as shown in Figure 5A–F, we found that silencing NOXA largely, though not completely, reversed the enhanced apoptosis triggered by these inhibitors in WSB2-deficient cells, suggesting that the sensitization effect is at least partially dependent on NOXA.

Regarding point (b): We have shown that WSB2 knockout alone had no impact on the levels of cleaved caspase-3/7/9 or the percentages of apoptotic cells in Huh-7 and C4-2B cells (Figure 4G-L and Figure 4—figure supplement 1A-D), indicating that WSB2 loss does not induce apoptosis on its own under basal conditions.

Regarding point (c): We appreciate the reviewer’s suggestion and have now repeated the experiment in WSB2 knockout Huh-7 cells. The new results further support the synthetic lethality between WSB2 loss and AZD5991 treatment (Figure 4—figure supplement 1C, D).

(6) Figure 5A/C/E. The effect of siNOXA is minor, if any, for cleavage of caspases. The same thing for Figure 6F/H.

We appreciate the reviewer’s insightful observation regarding the relatively modest effect of shNOXA on caspase cleavage in Figures 5A/C/E and Figures 6F/H. Indeed, we acknowledge that the reduction in caspase cleavage following NOXA knockdown is moderate. However, consistent with our discussions in the manuscript, NOXA knockdown significantly—but not completely—rescued the increased apoptosis observed in WSB2-deficient cells treated with BCL-2 family inhibitors. This suggests that while NOXA plays a notable role, additional mechanisms or unidentified targets may also be involved in WSB2-mediated regulation of apoptosis.

(7) Figure 5 I&J. The authors may consider performing IHC staining, immunofluorescence, or WB analysis to show the levels of NOXA and cleaved caspases or PARP in xenograft tumors. This would provide in vivo evidence of significant apoptosis induction resulting from the co-administration of ABT-737 and R8-C-terminal NOXA peptide.

We appreciate the reviewer's thoughtful suggestion regarding additional immunohistochemical or immunofluorescence analyses in xenograft tumors. However, due to current limitations in available antibodies suitable for reliable detection of NOXA by IHC and IF, we are unable to perform these experiments. We greatly appreciate the reviewer's understanding of this technical constraint. Nevertheless, our existing data collectively supports the conclusion that the combination of ABT-737 and R8-C-terminal NOXA peptide significantly enhances apoptosis in vivo.

(8) Figure 7. Does an inverse correlation exist between the protein levels of WSB2 and NOXA in RPAD or LIHC tissue microarrays? On page 12, in the first paragraph, Figure 7M-P was cited incorrectly.

We sincerely thank the reviewer for raising this important issue. As mentioned above, due to current limitations regarding the availability of suitable antibodies that can reliably detect NOXA by IHC, we regret that it is not feasible to experimentally address this question at this time.

Additionally, we have carefully corrected the citation error involving Figure 7M-P on page 12, as pointed out by the reviewer.

(9) Figure S1D. BCL-W levels were reduced upon WSB2 overexpression, which should be acknowledged.

We sincerely thank the reviewer for raising this important issue. We acknowledge that BCL-W protein levels were slightly reduced upon WSB2 overexpression in Figure S1D. However, this effect is distinct from the pronounced reduction observed in NOXA protein levels. We have revised the manuscript to clarify this point. Additionally, we recognize that transient overexpression systems may occasionally lead to non-specific or artifactual changes. Our exogenous expression and co-immunoprecipitation experiments did not support an interaction between BCL-W and WSB2. Therefore, the observed reduction of BCL-W under these conditions may not reflect a physiologically relevant regulation.

(10) Figure S4. Given WSB2 KO mice are viable; the authors may consider determining whether these mice are more sensitive to radiation-induced tissue damage or but more resistant to radiation-induced tumorigenesis?

We sincerely thank the reviewer for this insightful and biologically meaningful suggestion. We agree that investigating the potential role of WSB2 in radiation-induced tissue damage and tumorigenesis would be of great interest. However, conducting such experiments requires access to specialized irradiation facilities, which are currently unavailable to us. Nevertheless, we recognize the value of this line of investigation and plan to explore it in our future studies.

(11) All data were displayed as mean{plus minus}SD. However, for data from three independent experiments, it is more appropriate to present the results as mean{plus minus}SEM, not mean{plus minus}SD.

We sincerely thank the reviewer for highlighting this important issue. In line with the reviewer's suggestion, we have revised the manuscript accordingly and now present data from three independent experiments as mean ± SEM.

(12) The figure legends require careful review: (i) The low dose of ABT-199 (Figure 6H) and the dose of ABT-199 used in Figure 6I are missing. (ii) The legends for Figure S1D-E are incorrect. (iii) The name of the antibody in the legend of Figure S3C is incorrect.

We sincerely thank the reviewer for raising these important issues. We have carefully corrected all the errors mentioned. In addition, we have thoroughly reviewed the manuscript to prevent similar errors.

**Reviewer #2 (Recommendations For The Authors):**
The authors focus on NOXA, after initially identifying WSB2 to interact with several BCL2 proteins. The rationale behind this is that WSB2 depletion or overexpression affects NOXA levels, but none of the other BCL2 proteins tested, as stated in the text. Yet, BCLW is also depleted upon overexpression of WSB2 (Supplementary Figure 1). How does this phenomenon relate to the sensitization noted, is BCL-W higher in WSB2 KO cells? It does not seem so though. This warrants discussion.

We appreciate the reviewer for raising this important issue. Our results showed that overexpression of WSB2 markedly reduced NOXA levels, while the levels of other BCL-2 family proteins remained unaffected or minimally affected, such as BCL-W (Figure 2—figure supplement 1A). Furthermore, depletion of WSB2 through shRNA-mediated KD or CRISPR/Cas9-mediated KO in C4-2B cells or Huh-7 cells led to a marked increase in the steady-state levels of endogenous NOXA, without affecting other BCL-2 family proteins examined, included BCL-W (Figure 2A-C, Figure 2—figure supplement 2A, B).

If WSB2 depletion does not affect MCL1 levels, how does excess NOXA actually kill? Does it bind to any (other) prosurvival proteins under conditions of WSB2 depletion? Is the MCL1 half-life changed?

We appreciate the reviewer for raising this important point. NOXA is a BH3-only protein known to promote apoptosis primarily by binding to and neutralizing anti-apoptotic BCL-2 family members, especially MCL-1, via its BH3 domain. It can inhibit MCL-1 either through competitive binding or by facilitating its ubiquitination and subsequent proteasomal degradation. In our system, the total protein levels of MCL-1 remained unchanged in *WSB2* knockout cells, suggesting that NOXA may not be promoting apoptosis through enhanced MCL-1 degradation. Instead, we speculate that the accumulation of NOXA in *WSB2*-deficient cells enhances apoptosis by sequestering MCL-1 through direct binding, thereby freeing pro-apoptotic effectors such as BAK and BAX. In line with our observations, Nakao et al. reported that deletion of the mitochondrial E3 ligase MARCH5 led to a pronounced increase in NOXA expression, while leaving MCL-1 protein levels unchanged in leukemia cell lines (Leukemia. 2023 ;37:1028-1038., PMID: 36973350).

Additionally, NOXA has been reported to interact with other anti-apoptotic proteins, including BCL-XL. It is therefore possible that under conditions of WSB2 depletion, excess NOXA may also bind to BCL-XL and relieve its inhibition of BAX/BAK, further contributing to apoptosis. Future experiments assessing NOXA binding partners in *WSB2*-deficient cells would help clarify this mechanism.

I think some initial insights into the mechanism underlying the sensitization would add a lot to this study. Is there a role of BFL1/A1 in any of these cell lines, as it can also rather selectively bind to NOXA and is sometimes deregulated in cancer?

We appreciate the reviewer for raising this important issue. While BFL1/A1 is indeed another anti-apoptotic BCL-2 family member that can selectively bind to NOXA and has been implicated in cancer, our study primarily focuses on the WSB2-NOXA axis. However, given its potential involvement in apoptosis regulation, it would be an interesting direction for future studies to explore whether BFL1/A1 contributes to NOXA-mediated sensitization in specific cellular contexts.

Otherwise, this is a very nice and convincing study.